# Robotic search for optimal cell culture in regenerative medicine

Genki N Kanda[1,2,3*†], Taku Tsuzuki[4†], Motoki Terada[1,5], Noriko Sakai[1,5], Naohiro Motozawa[1], Tomohiro Masuda[1,5], Mitsuhiro Nishida[1,5], Chihaya T Watanabe[4], Tatsuki Higashi[4], Shuhei A Horiguchi[4], Taku Kudo[3], Motohisa Kamei[3], Genshiro A Sunagawa[1,6], Kenji Matsukuma[3], Takeshi Sakurada[4], Yosuke Ozawa[4*], Masayo Takahashi[1,5,7], Koichi Takahashi[2,8,9*], Tohru Natsume[3,10*]

[1]Laboratory for Retinal Regeneration, RIKEN Center for Biosystems Dynamics Research, Kobe, Japan; [2]Laboratory for Biologically Inspired Computing, RIKEN Center for Biosystems Dynamics Research, Osaka, Japan; [3]Robotic Biology Institute Inc., Tokyo, Japan; [4]Epistra Inc., Tokyo, Japan; [5]VCCT Inc., Kobe, Japan; [6]Laboratory for Molecular Biology of Aging, RIKEN Center for Biosystems Dynamics Research, Kobe, Japan; [7]Vision Care Inc., Kobe, Japan; [8]Graduate School of Media and Governance, Keio University, Fujisawa, Japan; [9]Graduate School of Frontier Biosciences, Osaka University, Suita, Japan; [10]Department of Life Science and Biotechnology, Cellular and Molecular Biotechnology Research Institute, National Institute of Advanced Industrial Science and Technology, Tokyo, Japan

**\*For correspondence:**
genki.kanda@riken.jp (GNK);
ozawaysk@epistra.jp (YO);
ktakahashi@riken.jp (KT);
t-natsume@aist.go.jp (TN)

[†]These authors contributed equally to this work

**Abstract** Induced differentiation is one of the most experience- and skill-dependent experimental processes in regenerative medicine, and establishing optimal conditions often takes years. We developed a robotic AI system with a batch Bayesian optimization algorithm that autonomously induces the differentiation of induced pluripotent stem cell-derived retinal pigment epithelial (iPSC-RPE) cells. From 200 million possible parameter combinations, the system performed cell culture in 143 different conditions in 111 days, resulting in 88% better iPSC-RPE production than that obtained by the pre-optimized culture in terms of the pigmentation scores. Our work demonstrates that the use of autonomous robotic AI systems drastically accelerates systematic and unbiased exploration of experimental search space, suggesting immense use in medicine and research.

## Editor's evaluation

The manuscript by Kanda GN, Natsume T et al. describes a robotic artificial intelligence system with a batch Bayesian optimization algorithm that allows to optimise and reliably repeat cell culture protocols. The authors utilise induced pluripotent stem cell-derived retinal pigment epithelial cells as a model culture system of broad interest in regenerative medicine. They demonstrate that the robotic system with a Bayesian algorithm accelerates the optimisation of cell culture protocols and increases the quality and quantity of cell products, compared with manual operations – these results will likely inform and strongly impact modern cell culture strategies in regenerative medicine. The manuscript clearly explains the parameters analysed, the methods and analyses performed, current limitations and possible broader future use beyond the system tested.

## Introduction

Automating scientific discovery is one of the grandest challenges of the 21st century (*Kitano, 2021*; *Kitano, 2016*). A promising approach involves creating a closed loop of computation and

experimentation by combining AI and robotics (*King et al., 2009*). A relatively simple form of autonomous knowledge discovery involves searching for optimal experimental procedures and parameter sets through repeated experimentation and result validation, according to a predefined validation method. For example, in material science, the parameters associated with the growth of carbon nanotubes have been explored using an autonomous closed-loop learning system (*Nikolaev et al., 2016*). In experimental physics, Bayesian optimization has been used to identify the optimal evaporation ramp conditions for Bose–Einstein condensate production (*Wigley et al., 2016*). In 2019, a promoter-combination search in molecular biology was automated using an optimization algorithm-driven robotic system (*HamediRad et al., 2019*). Some robotic systems for cell culture have already been developed (*dos Santos et al., 2013*; *Kino-Oka et al., 2009*; *Konagaya et al., 2015*; *Liu et al., 2010*; *Matsumoto et al., 2019*; *Nishimura et al., 2019*; *Ochiai et al., 2021*; *Soares et al., 2014*; *Thomas et al., 2008*); however, many of these fixed-process automation apparatuses lack the flexibility and precision necessary to execute comprehensive parameter searching.

Here, we report the development of a robotic search system that autonomously and efficiently searches for the optimal conditions for inducing iPS cell differentiation into retinal pigment epithelial (RPE) cells (iPSC-RPE cells). The system replaces the manual operations involved in cell culture with robotic arms. Cell culture is probably one of the most delicate procedures in two respects. First, the parameters related to physical manipulation can greatly affect the outcome of the experiment (*Kanie et al., 2019*). Secondly, it takes a long time to execute a series of protocols. For example, cells artificially differentiated from embryonic stem cells or induced pluripotent stem cells (ES/iPS cells) need to be processed using hundreds of experimental procedures that typically last for weeks or months before they can be used for transplantation in regenerative medicine.

During these processes, cells are given chemical perturbations (e.g. type, dose, and timing of reagents) and physical perturbations (e.g. strength of pipetting, vibration during handling of plates, timing of transfer from/to $CO_2$ incubator, and accompanying changes in factors such as temperature, humidity, and $CO_2$ concentration). Due to the heterogeneous and complex internal states of cells, suitable culture conditions must be determined for each strain and/or lot (*Kino-Oka and Sakai, 2019*). A small difference in a single chemical stimulus or physical procedure can lead to failure of differentiation or poor quality of the produced cells, and such consequences can often become experimentally detectable only days or weeks after the input is given (*Kino-oka et al., 2019*). Therefore, the use of robotic arms is a great addition in the search for optimal cell culture conditions because robots can repeatedly perform the same operation with high precision. Moreover, they hardly make any errors, which are logged when committed.

It is advantageous to utilize high-accuracy and programmable robotic arms for the search of optimal cell culture parameters. Unlike human hands, robotic arms can repeatedly perform the same procedure. They ensure reproducibility by keeping all parameters related to physical procedures constant. Furthermore, the actual operations are logged by the software along with sensor information when they are deviated from the established programs. Thus, robotization provides an ideal parameterization of experimental procedures. Some automated cell culture machines have already been proposed (*Regent et al., 2019*); however, proper formulation of an autonomous search for optimal culture conditions has not yet been determined.

In this study, we combined a Maholo LabDroid (*Yachie et al., 2017*) and an AI system that independently evaluates the experimental results and plans the next experiments to realize an autonomous robotic search for optimal culture conditions. We first created a digital representation of the regenerative medical cell culture protocol used for iPS cell differentiation into retinal pigment epithelial (RPE) cells (iPSC-RPE cells) (*Mandai et al., 2017*), which can be executed by the robot and used as a template for an AI-driven parameter search (*Figure 2—video 1*). We then implemented the experimental protocol on a LabDroid, which is a versatile humanoid robot that can perform a broad range of experimental procedures. Its flexibility allows frequent changes in protocols and protocol parameters, making it suitable for use in experimental parameter searches. The robot has an integrated microscope that provides data for image-processing through AI, which evaluates the quality of growing cells. The search process was mathematically formulated as a type of experimental design problem, and a batch Bayesian optimization (BBO; *Figure 1*, *Figure 1—figure supplements 1 and 2*) technique was employed as a solver. Finally, we demonstrated that iPSC-RPE cells generated by LabDroid satisfy the cell biological criteria for regenerative medicine research applications.

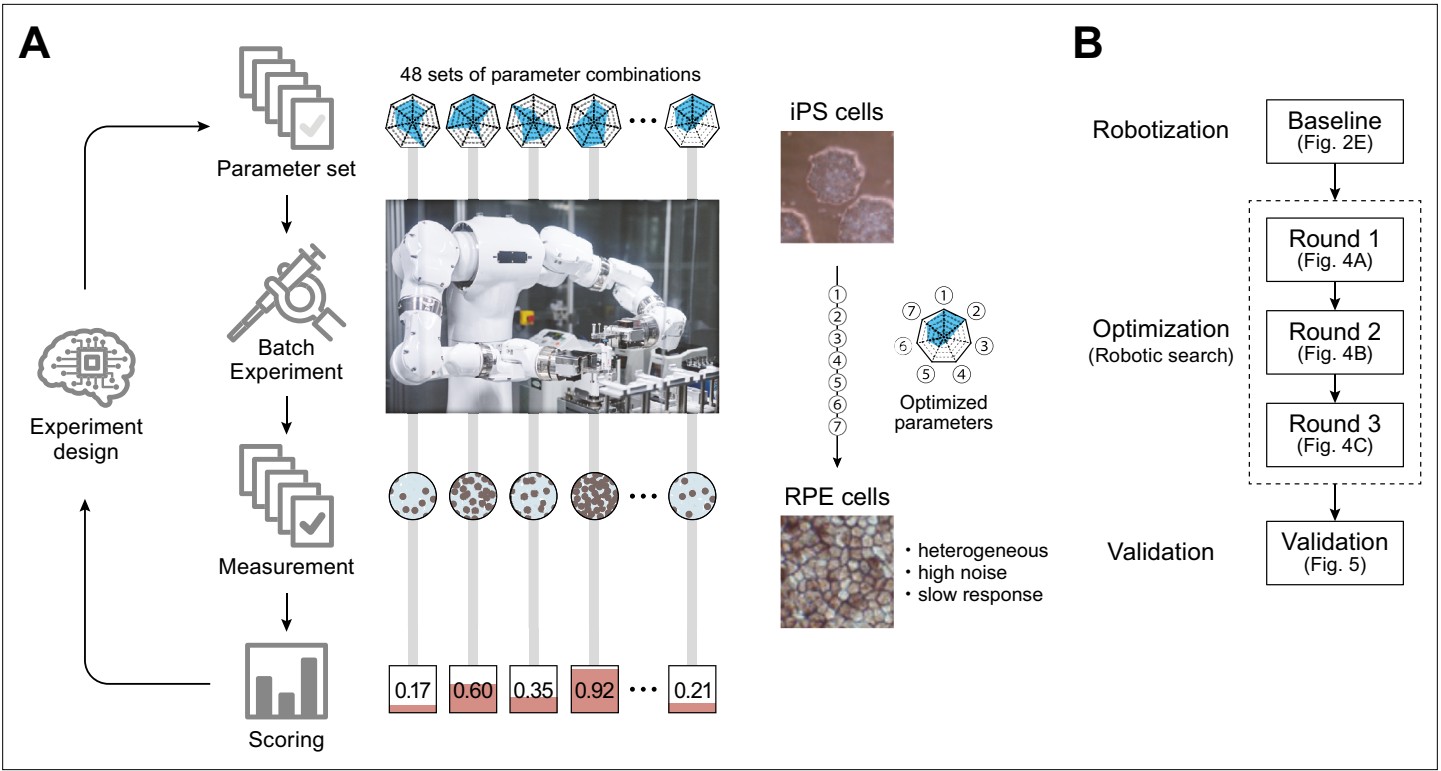

**Figure 1.** Robotic search for optimal experimental conditions. (**A**) Overall workflow for the optimization of experimental procedures using combined experimental robotics and Bayesian optimization. The user defines the target experimental protocol, subject parameters of the protocol, and the validation function. In this study, we chose the differentiation procedure from iPS to RPE cells as a target protocol and selected the reagent concentration, administration period, and five other parameters (details are shown in *Table 1*). We defined the pigmented area in a culture well, which represents the degree of RPE differentiation induction, as the validation function. The optimization program presented multiple parameter candidates; the LabDroid performed the experiment, and then an evaluation value for each candidate was obtained. Subsequently, the Bayesian optimization presented a plurality of parameter candidates predicted to produce higher validation values. The optimal parameters were searched by repeating candidate presentation, experiment execution, validation, and prediction. The detailed components are shown in *Figure 1—figure supplement 2*. (**B**) Workflows performed in this study. First, robotization of the iPSC-RPE protocol was performed as a baseline. Next, the optimization process was conducted in three rounds, followed by statistical and biological validation. The figure numbers in parentheses represent the results shown in the figure.

The online version of this article includes the following figure supplement(s) for figure 1:

**Figure supplement 1.** Schematic diagram of iPSC-RPE transplantation.

**Figure supplement 2.** System components.

## Results

### Robotization of the iPSC-RPE differentiation protocol

An overview of the iPSC-RPE differentiation protocol used for optimization is shown in *Figure 2A* and *Figure 1—figure supplement 1*. It consists of five steps: seeding, preconditioning, passage, RPE differentiation (induction), and RPE maintenance culture. The day on which the passage was performed was defined as differentiation day (DDay) 0, and the cultured cells were sampled and validated on DDays 33 and 34. To implement this protocol using LabDroid, the necessary peripheral devices were installed on and around LabDroid's workbench (*Figure 2B*, *Figure 2—figure supplement 1*). We designed the system to work simultaneously with eight 6-well plates per batch, for a total of 48 cell-containing wells. LabDroid was programmed for three types of operations: seeding, medium exchange, and passage (*Figure 2—figure supplements 2–7*; *Figure 2—source data 3*; *Figure 2—video 1*). The steps for the preconditioning and induction, which correspond to the preparation of reagents, were named medium exchange type I, and the step for RPE maintenance culture, which does not involve reagent preparation, was named medium exchange type II (*Figure 2A*, *Figure 2—figure supplement 2*).

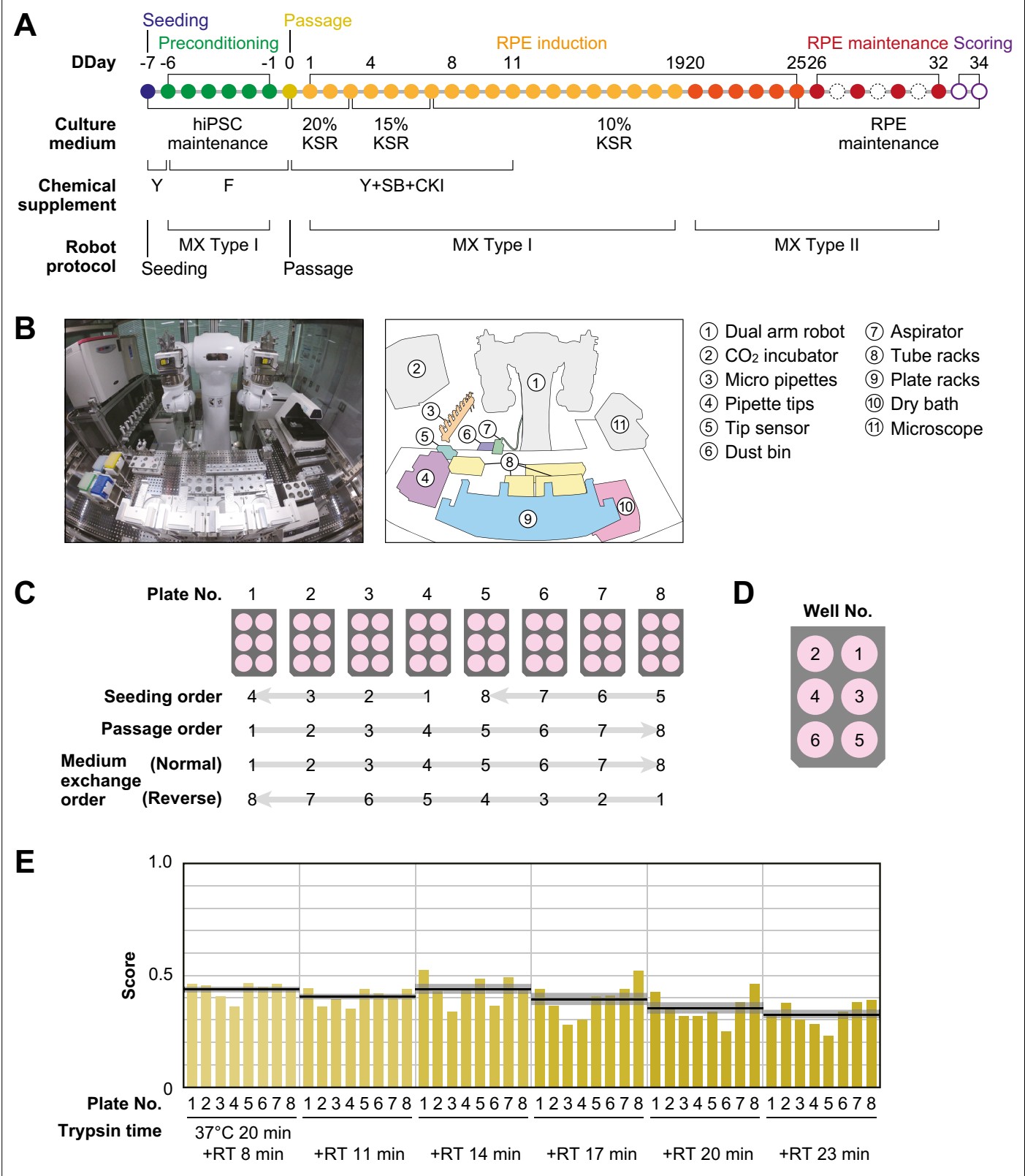

**Figure 2.** Robotization of iPSC-RPE differentiation protocols. (**A**) Schematic diagram of the standard iPSC-RPE differentiation procedures. DDay indicates the differentiation day. Filled circles represent days when the robot operated, solid circles represent days with human operations only, and dashed line circles represent days when no operations were conducted. F stands for FGF receptor inhibitor; Y for Y-27632, a Rho-kinase inhibitor; SB for SB431542, a TGF-β/Activin/Nodal signal inhibitor; CKI for a CKI-7, Wnt signal inhibitor; and MX for medium exchange. (**B**) The LabDroid Maholo

*Figure 2 continued on next page*

*Figure 2 continued*

including peripheral equipment. (**C**) Plate numbering and the orders of seeding, passage, and medium exchange operations. Eight 6-well plates were used for each experiment. (**D**) Well numbering. (**E**) Scores of the first trial. iPSC-RPE differentiation was conducted under six different trypsin treatment times using the LabDroid. Yellow bars represent the pigmented cell area score of each well. The bold black lines and the shaded area around the lines represent the mean score and SEM of eight samples operated at the same trypsin time, respectively. The raw values are shown in *Figure 2—source data 2*.

The online version of this article includes the following video, source data, and figure supplement(s) for figure 2:

**Source data 1.** Acquired pigmented images of the baseline experiment.

**Source data 2.** Executed parameters and scores of the baseline experiment.

**Source data 3.** Pipetting volume and pipette combination.

**Figure supplement 1.** LabDroid Maholo booth.

**Figure supplement 2.** Workflows of the experimental operation.

**Figure supplement 3.** Representative LabDroid execution of a seeding experiment (round 3, DDay −7).

**Figure supplement 4.** Representative LabDroid execution of a preconditioning experiment (round 3, DDay −6, 1st run).

**Figure supplement 5.** Representative LabDroid execution of a passage experiment (round 3, DDay 0, 1st run).

**Figure supplement 6.** Representative LabDroid execution of an RPE differentiation experiment (round 3, DDay 10, 1st run).

**Figure supplement 7.** Representative LabDroid execution of an RPE maintenance experiment (round 3, DDay 32, 1st run).

**Figure supplement 8.** Image processing to calculate the pigmentation scores.

**Figure 2—video 1.** Representative LabDroid movements.

https://elifesciences.org/articles/77007/figures#fig2video1

First, we used LabDroid to perform baseline experiments involving the induction of iPSC-RPE cell differentiation under the same conditions as the typical manual operations. Because of the differences in structure and experimental environment between the LabDroid and humans, some operations and movements, such as the use of a centrifuge, the presence or absence of cell counting at the time of passage, and the speed of movement, differed from those of humans. For example, achieving the same time interval for trypsin treatment in all wells of a single plate during cell detachment using LabDroid is difficult. Therefore, the passage operation was performed at six separate time intervals. The cells differentiating into RPE cells produce melanin, which causes them to turn brown. Therefore, the area ratio of the total number of pigmented cells on DDay 34 was used to estimate the differentiation induction efficiency and obtain evaluation scores, following the example of previous studies (*Kuroda et al., 2019*; *Regent et al., 2019*; *Figure 2—figure supplement 8*). These validation scores were used to simplify the validation process and do not reflect the entire quality of the RPE.

Baseline experiments were conducted and validated using six trypsin conditions and eight plates (*Figure 2C–E*; *Figure 2—source data 1* and *Figure 2—source data 2*). The highest scoring was obtained when trypsin treatment was conducted for 20 min at 37 °C, followed by 14 min incubation at room temperature (RT, approximately 25 °C), with an eight-plate score of 0.44±0.03 (mean ± SEM, n=8). The lowest scoring was obtained when trypsin treatment was conducted for 20 min at 37 °C, followed by 23 min at RT, with an eight-plate score of 0.33±0.02 (mean ± SEM, n=8). LabDroid successfully performed the iPSC-RPE protocol, as evidenced by the detection of pigmented cells in all 48 wells and the lack of errors in the operating process. However, in the naive transplantation of the manual protocol to the robot, the induction efficiency was insufficient. This suggests that it is inherently difficult to describe physical parameters, including unrecorded human movements. Therefore, we attempted to optimize the protocol parameters to further improve the scores using a robotic search.

## Parameterization of the protocol

To improve the pigmentation score, we selected seven parameters for optimization: two from the preconditioning step, three from the passage step, and two from the induction step. Search domains were set for each parameter (*Table 1*; *Figure 3A and B*).

From the preconditioning step on DDays −1 to −6, we selected two parameters for optimization: the concentration of fibroblast growth factor receptor inhibitor (FGFRi) in the medium (*PC*, preconditioning concentration), and the duration of addition (*PP*, preconditioning period). From the passage

**Table 1.** Definition of optimized parameters.

Parameter names, parameter name codes, description, parameter ranges, parameter units, correspondence between experimental procedure and parameters used (related to *Figures 2A, 3A and B*).

| Parameter name | Code | Description | Range | Unit | Protocol step |
|---|---|---|---|---|---|
| Preconditioning concentration | *PC* | FGFRi concentration in medium | 0–505 | nM | Preconditioning |
| Preconditioning period | *PP* | FGFRi duration in medium | 1–6 | day | Preconditioning |
| Detachment trypsin period | *DP* | Trypsin incubation duration at room temperature after incubation at 37 °C, 20 min. | 5, 8, 11, 14, 17, 20, 23 | min | Passage |
| Detachment pipetting strength | *DS* | Pipetting strength during cell detachment | 10–100 | mm/s | Passage |
| Detachment pipetting length | *DL* | Bottom surface area to be pipetted | short / long | N/A | Passage |
| KSR period | *KP* | KSR concentration and duration in medium: KSR concentration is decreased linearly every day so that KSR becomes 10% on DDday of KP value | 1–19 | day | RPE differentiation |
| Three supplements period | *3P* | Three chemical supplements duration | 3–19 | day | RPE differentiation |

step performed on DDay 0, we selected three parameters to optimize: the pipetting strength during cell detachment (*DS*, detachment pipetting strength), the area of the bottom surface to be pipetted (*DL*, detachment pipetting length), and trypsin processing time (*DP*, detachment trypsin period) of a passage. *DP* is a contextual parameter that can only be used to perform experiments at fixed values, owing to the specifications of the experimental system. In this case, *DP* is allowed to take different fixed values at three-minute intervals, corresponding to the number of wells in the plate. From the induction step on DDays 1–25, we selected two parameters to optimize: the concentration of KnockOut Serum Replacement (KSR) in the medium (*KP*, KSR period), and the duration of exposure period of the three chemical supplements (*3P*, three supplement period).

## Optimization of the protocol

To improve the optimization performance, 48 conditions (eight plates × six wells, as shown in *Figure 2C*) were executed in parallel in each batch. The 48 conditions were selected from the search space using the Bayesian optimization module to maximize the acquisition function calculated from the past experimental data. In general, solving a high-dimensional, expensive black-box optimization problem such as the present one with a limited number of rounds is challenging. In our case, some 200 million possible parameter combinations existed in the search space, and the point where the pigmented score was optimal in three rounds (144 queries) had to be determined, because one experiment round took 40–45 days. In recent studies, BBO has shown excellent performance in real-world black-box optimization problems (*Burger et al., 2020*; *Gongora et al., 2020*; *HamediRad et al., 2019*). We integrated an experimental design module based on BBO to effectively search for the optimal experimental parameters that maximize the pigmentation scores in the search space defined in *Figure 3B*.

The Bayesian optimization module generates queries using two components: the Model updater, which updates the surrogate model that captures the relationship between parameters and the scores using Bayesian inference (*Figure 3—figure supplement 1*); the Query generator, which generates the next experimental parameters $X_{next}$ using an acquisition function and a policy function (*Figure 3C*, *Figure 3—figure supplement 2*; Algorithm 1–3). In the Query generator, the acquisition function estimates the expected progress toward the optimal experimental parameter at a given experimental parameter (*Figure 3D*). Then, using the acquisition function, the policy function generates the next 48 experimental parameters $X_{next}$ considering the context of trypsin processing time $x_{DP}$ (*Figure 3E*).

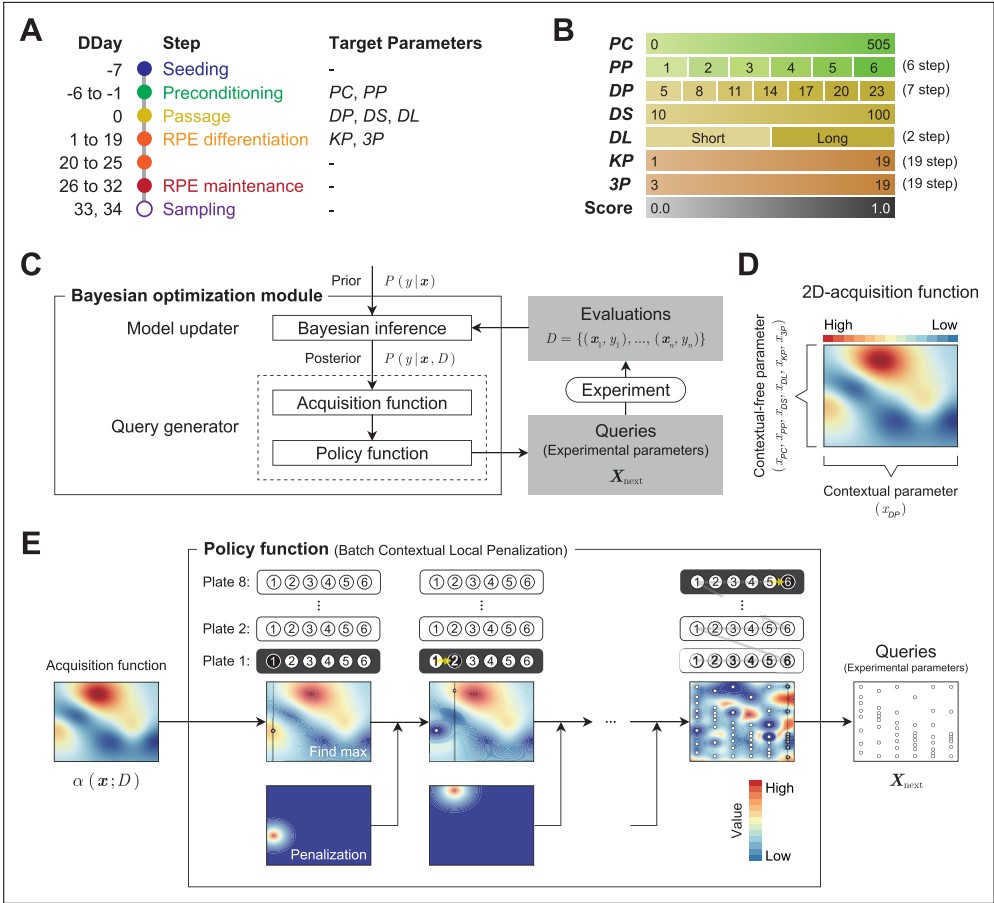

**Figure 3.** Optimization module. (**A**) Definition of the target parameters and corresponding steps in the protocol: *PC*, preconditioning concentration; *PP*, preconditioning period; *DP*, detachment trypsin period; *DS*, detachment pipetting strength; *DL*, detachment pipetting length; *KP*, KSR concentration reducing period; and *3P*, three chemical (Y, SB, CKI) supplement administration period. (**B**) Ranges and stepping of the parameters. (**C**) The Bayesian optimization module consists of two components: a Model updater and a Query generator. The Model updater updates the Gaussian process posterior on the experiment using all available data $\mathcal{D} = \left\{ (x_i, y_i) \right\}_{i=1}^{n}$, where x indicates experimental parameter, and y indicates corresponding evaluation score. The Query generator calculates the acquisition function $\alpha(x; \mathcal{D})$ for an experiment parameter $x$ with the posterior distribution $P(y|x, \mathcal{D})$, and generates the experiment parameter set $X_{\text{next}}$ for the next 48 points using the policy function with $\alpha(x; \mathcal{D})$. (**D and E**) Test of the query generation process using a two-dimensional toy acquisition function. (**D**) Values of the toy acquisition function given an experimental parameter set. The horizontal axis represents the input values of $x_{DP}$ (contextual parameter), whereas the vertical axis represents the input values of the other six remaining context-free parameters $X = (x_{PC}, x_{PP}, x_{DS}, x_{DL}, x_{KP}, x_{3P})$, which are collapsed into a single axis. The color of the heatmap indicates the value of the acquisition function. In the heat map, the acquisition value is higher in places where the color is closer to red and lower in places where the color is closer to blue. (**E**) Test of the query generation process for the experimental parameter set $X_{\text{next}}$ in the next experiment using a batch contextual local penalization policy (BCLP). The heat maps in the upper row show the (penalized) acquisition function values, and the lower row shows the penalization values for the acquisition function. The queries $X_{\text{next}}$ for 48 wells (right side figure) were iteratively generated from the maximization-penalization loop on the acquisition function.

The online version of this article includes the following figure supplement(s) for figure 3:

**Figure supplement 1.** Demonstration of how Gaussian process regression updates a Bayesian posterior.

**Figure supplement 2.** Demonstration of Bayesian optimization generation of experimental parameters.

**Figure supplement 3.** Toy testing function based on domain knowledge.

**Figure supplement 4.** Preliminary testing of the Bayesian optimization under different conditions.

---

**Algorithm 1. Batch Bayesian Optimization for iPSC-RPE differentiation protocol.**

---

Input: The search space $\chi$, GP prior $(\mu_0, \sigma_0, k)$, number of rounds $M$, number of Plates $P$, number of Wells $W$, Dataset $\mathcal{D} = \{(x_i, y_i)\}_{i=1}^{n}$

for $t=1$ to $M$ do

 1. Construct GP posterior $(\mu_t, \sigma_t, k)$ using $\mathcal{D}$.
 2. Get the acquisition function $\alpha(x; \mathcal{D})$.
 3. Generate a experiment parameter set $X_{\text{next}}$ using the policy function.
 Execute the experiments $f(X_{\text{next}})$ .
 Append the experiment results to past data $\mathcal{D} = \mathcal{D} \cup \{(X_{\text{next}}, f(X_{\text{next}}))\}$.
 4. Compute optimal context $c_{DP}$ on Detach trypsin Period in the next experiment.

end

---

**Algorithm 2. The policy function for the iPSC-RPE differentiation protocol.**

---

Input: The acquisition function $\alpha(x; \mathcal{D})$, number of Plates $P$, number of Wells $W$

Output: The next experiment parameter set $X_{\text{next}} = \{(x_{t,p,w})\}_{(p,w)=1}^{(P,W)}$

1. Calculate utility functions from the acquisition function
 $\tilde{\alpha}_0(x; \mathcal{D}) \leftarrow g(\alpha(x; \mathcal{D}))$

 $\tilde{\alpha}(x; \mathcal{D}) \leftarrow \tilde{\alpha}_0(x; \mathcal{D})$
2. Generate next experiment parameters $X_{\text{next}} = \{(x_{t,p,w})\}_{(p,w)=1}^{(P,W)}$ in Maximization-Penalization loop

for $P=1$ to $P$ do
 for $w=1$ to $W$ do
 1. maximization-step: $x_{t,p,w} \leftarrow \text{argmax}_{x \in \chi} \{\tilde{\alpha}(x; \mathcal{D})\}$
 2. penalization-step: $\tilde{\alpha}(x; \mathcal{D}) \leftarrow \tilde{\alpha}_0(x; \mathcal{D}) \prod_{(k,h)=1}^{(p,w)} \varphi(x; x_{t,k,h}, \hat{L})$
 end
end

---

**Algorithm 3. Detachment trypsin period adjustment on the iPSC-RPE differentiation protocol.**

---

Input: The acquisition function $\alpha(x; \mathcal{D})$, current $DP$ context $c_{DP,t}$, context shift width $\Delta c$
Output: The next $DP$ context $c_{DP, t+1}$
1. Candidates of $DP$ context ranges for the next round. (In this study, $\Delta c = 3$ min)
$c_{DP} \leftarrow c_{DP, t}$

$c_{DP}^{-} \leftarrow c_{DP, t} - \Delta c$

$c_{DP}^{+} \leftarrow c_{DP, t} + \Delta c$
2. Calculate values $V$, $V^{-}$, $V^{+}$ that accumulate $\alpha(x; \mathcal{D})$ on each context ranges $c_{DP}, c_{DP}^{-}, c_{DP}^{+}$
$V = \sum_i \int_\chi \alpha(x; \mathcal{D}, x_{DP} = c_{DP, i})$
$V^{-} = \sum_i \int_\chi \alpha(x; \mathcal{D}, x_{DP} = c_{DP, i}^{-})$
$V^{+} = \sum_i \int_\chi \alpha(x; \mathcal{D}, x_{DP} = c_{DP, i}^{+})$
3. Calculate ratios $R^{-}$, $R^{+}$ between each values defined above.
$R^{-} = V^{-}/V$

$R^{+} = V^{+}/V$
4. Choose the next $DP$ context $c_{DP, t+1}$ in following rules.
if $(\max R^{-}, R^{+} < 1.05)$ then
 $c_{DP, t+1} \leftarrow c_{DP}$
end
else if $(R^{-} > R^{+})$ then
 $c_{DP, t+1} \leftarrow c_{DP}^{-}$
end
else if $(R^{-} \leq R^{+})$ then
 $c_{DP, t+1} \leftarrow c_{DP}^{+}$
end

---

 To test the performance of the Bayesian optimization module in our case, we executed a preliminary performance validation using a toy testing function constructed on domain knowledge (*Figure 3— figure supplements 3 and 4*).

## Robotic optimization drastically improved the pigmentation score

In this study, three successive experiments were conducted to optimize the target protocol. In each round, 48 conditions were generated using the Bayesian optimization module and translated into LabDroid operating programs. The robot performed 40 days of iPSC-RPE induction culture under each

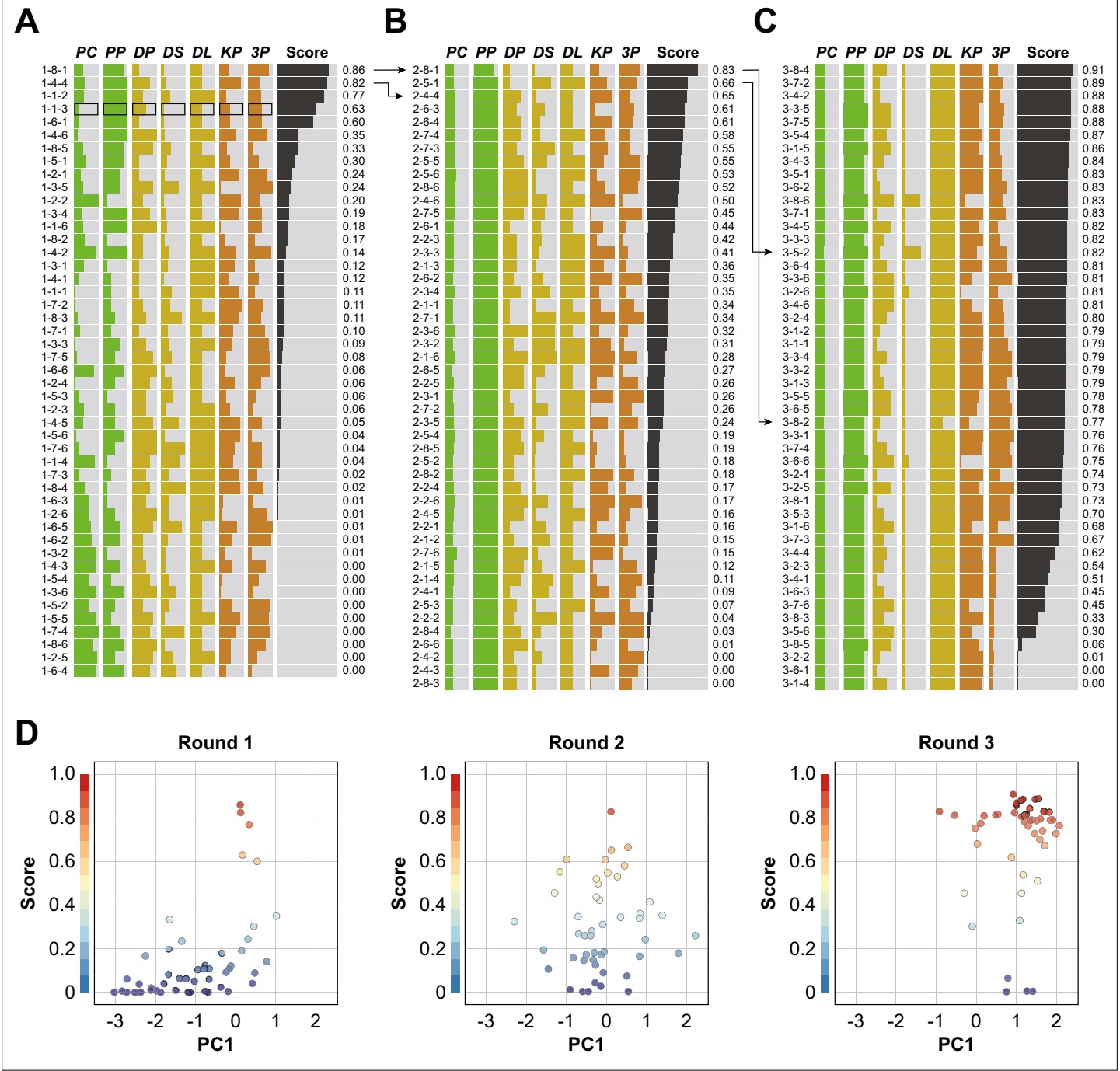

**Figure 4.** Robotic search for optimal parameters in iPSC-RPE differentiation. (**A–C**) Parameter candidates sorted in order of the pigmentation score in optimization rounds 1 (**A**), 2 (**B**), and 3 (**C**). The ID label on the left represents 'Round No. - Plate No. - Well No.'. For example, '1-2-3' means '(Round) 1-(Plate) 2-(Well) 3'. The parameter values and resulting pigmentation scores are plotted as horizontal bars. The parameter candidate with black frames (1-1-3) in (**A**) is the standard condition. Arrows indicate the control experiments; the top two conditions in round 1 were included in round 2, and the top two conditions in round 2 were implemented in round 3. The raw values are shown in *Figure 4—source data 4*. (**D**) Visualization of the parameter set and the pigmentation score distributions using partial least squares regression (PLS) in each round. The horizontal axis PC1 shows the values of the parameter candidates that are projected onto the first component of the PLS. The vertical axis shows the pigmentation score for each candidate parameter. As the rounds progressed, the overall score tended to converge in a higher direction. A full visualization of the experimental results using a parallel coordinate plot (PCP) is shown in *Figure 4—figure supplement 1*.

The online version of this article includes the following video, source data, and figure supplement(s) for figure 4:

**Source data 1.** Acquired pigmented images of the round 1 experiment.

*Figure 4 continued on next page*

*Figure 4 continued*

**Source data 2.** Acquired pigmented images of the round 2 experiment.

**Source data 3.** Acquired pigmented images of the round 3 experiment.

**Source data 4.** Executed parameters and scores of the optimization experiments.

**Figure supplement 1.** Parallel coordinate plot (PCP) of the robotic search experimental results.

**Figure 4—video 1.** Seeding operation.

https://elifesciences.org/articles/77007/figures#fig4video1

**Figure 4—video 2.** Preconditioning operation.

https://elifesciences.org/articles/77007/figures#fig4video2

**Figure 4—video 3.** Passaging operation.

https://elifesciences.org/articles/77007/figures#fig4video3

**Figure 4—video 4.** RPE differentiation operation.

https://elifesciences.org/articles/77007/figures#fig4video4

**Figure 4—video 5.** RPE maintenance operation.

https://elifesciences.org/articles/77007/figures#fig4video5

condition, and we obtained the rate of pigmented cells in the dish as an evaluation score (pigmentation score) for each condition. In accordance with the experimental design, we incorporated the two highest-scoring conditions from the previous experiment (*Figure 2E*) as control conditions, performed differentiation-inducing cultures with the LabDroid, and validated the area of the colored cells. In round 1, although one condition was found to be experimentally deficient, the other 47 conditions were validated. The highest score was 0.86 (*Figure 4A*; *Figure 4—source data 1*, *Figure 4—source data 4*), yielding five conditions that exceeded the mean value (0.39) for all wells in the baseline experiment (*Figure 2E*). In round 2, 46 conditions were generated, and the two highest-scoring conditions in round 1 were incorporated as control conditions. The highest score was 0.83 (*Figure 4B*; *Figure 4—source data 2*, *Figure 4—source data 4*). In round 3, 48 experiments were conducted, yielding an improved highest score of 0.91. We obtained 26 other conditions that were better than the highest in round 2 (*Figure 4C*; *Figure 4—source data 3*, *Figure 4—source data 4*). A visualization diagram of a two-dimensional partial least squares regression (PLS) clearly revealed that the overall experimental parameters tended to converge in a higher pigmented score direction from rounds 1 to 3 (*Figure 4D*, *Figure 4—figure supplement 1*).

To determine whether the optimized conditions were statistically improved over the pre-optimized conditions, an additional multi-well validation experiment was conducted after round 3 using the top five conditions in round 3 and the pre-optimized conditions. The validation values, ordered by place, were 0.71±0.06, 0.72±0.03, 0.76±0.02, 0.79±0.02, and 0.81±0.02 (mean ± SEM, n=3 each). All scores after optimization were statistically significantly higher than the pre-optimization scores (0.43±0.02; mean ± SEM, n=3) (*Figure 5A and B*; *Figure 5—source data 1*, *Figure 5—source data 2*).

In summary, we conducted 216 40-day cell culture experiments with a total experimentation time of 8640 days. We accelerated the search using a BBO technique, compressing the search time to 185 days with a cumulative robot operating time of 995 hr (*Figure 5—source data 5*; *Figure 5—figure supplements 1 and 2*; *Figure 4—videos 1–5*).

In this study, we succeeded in replacing part of the process of iPS cell differentiation into RPE cells for transplantation using robots, and demonstrated an effective optimization method (*Figure 1—figure supplement 2*). However, it was unclear whether robot-manufactured RPE cells would have the characteristics required for transplantation. Therefore, we purified the cells of the validation round, prepared them for transplantation, and performed a biological quality evaluation (*Figure 1—figure supplement 1B*). The analyzed iPSC-RPE cells expressed *BEST1*, *RPE65*, and *CRALBP* (*Figure 5C*), which are characteristic marker genes of RPE cells. In addition, we observed secretion of VEGF and PEDF into the culture medium, a characteristic of RPE cells (*Figure 5D and E*; *Figure 5—source data 3*). The expression of tight junction-associated factor ZO-1 was examined using immunohistochemistry, and a ZO-1-derived fluorescence signal was observed in microphthalmia-associated transcription factor (MITF)-positive cells, which play a central role in RPE cell function (*Figure 5F*). These results indicated that the robot-manufactured iPSC-RPE cells had the characteristics of RPE cells, and fulfilled

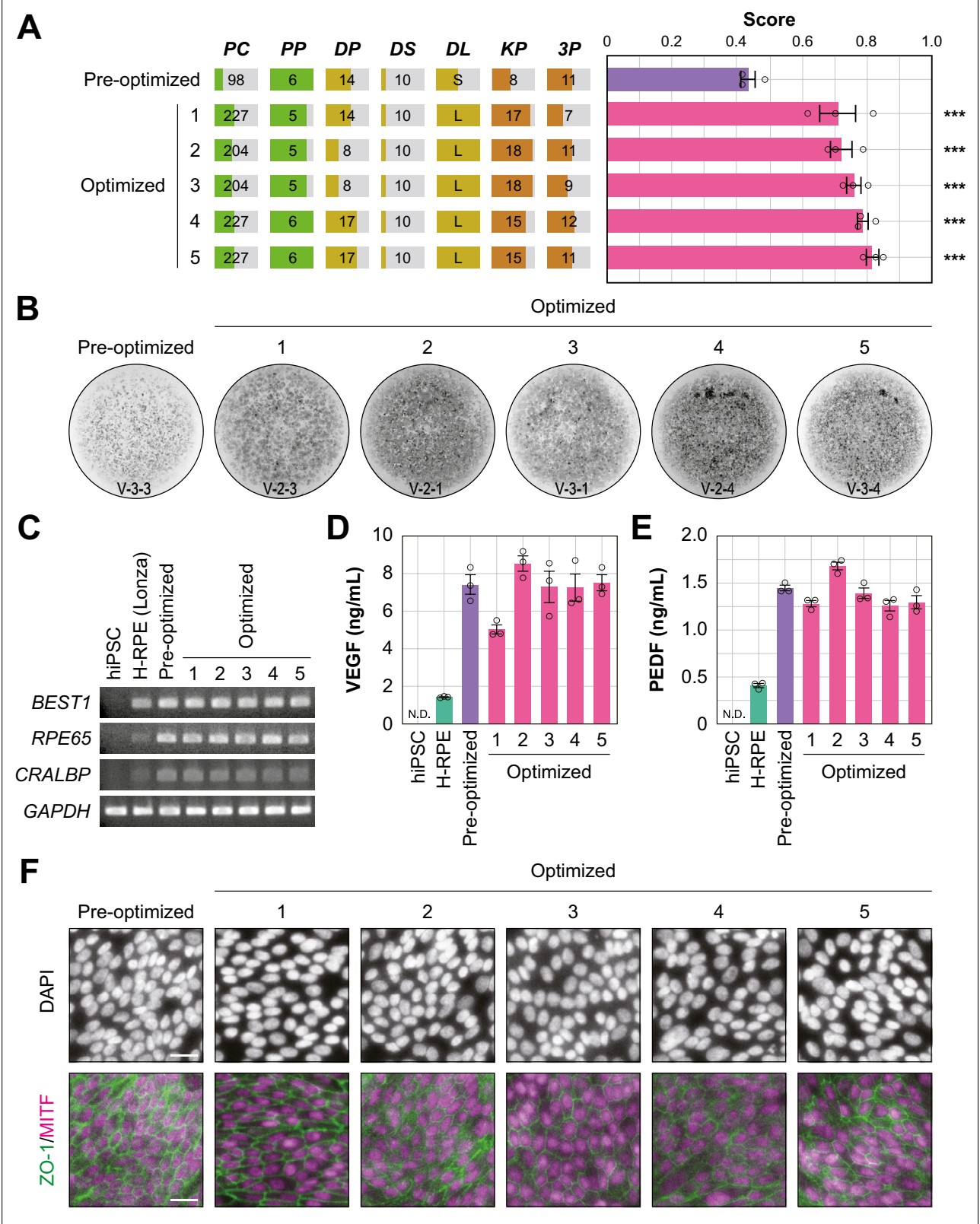

**Figure 5.** Quality evaluation of robot-induced RPE cells. (**A**) The pigmentation score evaluation of the pre-optimized conditions (n=3) and the top five conditions from round 3. Error bars represent the standard error of the mean (SEM). The numbers 1–5 in the optimized group represent the first to fifth place conditions for round 3 (*Figure 4C*). Circles represent an individual score, bars represent the mean score, and error bars represent the SEM. Statistical significance was examined using two-way ANOVA and SNK *post-hoc* tests. p<0.05 was considered significant. ***p<0.001 versus pre-

*Figure 5 continued on next page*

*Figure 5 continued*

optimized. In all other combinations, no statistical significance was detected. Raw values are shown in *Figure 5—source data 2*. (**B**) Representative pigmented images of the pre-optimized and five optimized iPSC-RPE cells. Images acquired on DDay 34. ID labeling on the bottom reads 'V (validation) - Plate No. - Well No.'. The other images are shown in *Figure 5—source data 1*. (**C–F**) Cell biological validation of the robot-induced RPE cells. After DDay 34, cells were purified, stocked, initiated, maintained for four weeks, and analyzed (*Figure 1—figure supplement 1B*). (**C**) Representative marker gene expression in RPE cells by RT-PCR. iPSC, undifferentiated iPSC; H-RPE (Lonza), Clonetics H-RPE (Lot #493461, Lonza, USA); pre-optimized and optimized LabDroid-induced RPE. (**D–E**) Quantification of representative secreted proteins from iPSC-RPE cells using ELISA. The supernatants were collected and the amount of VEGF (**D**) and PEDF (**E**) in the culture medium was analyzed 24 hr after medium exchange (n=3 wells each). Circles represent individual scores, bars represent the mean score, and error bars represent SEM. n.d.=not detected. The raw values are shown in *Figure 5—source data 3*. (**F**) Co-staining of ZO-1 (green) and MITF (magenta) using immunohistochemistry. Nuclei were stained with DAPI. The scale bars represent 20 μm.

The online version of this article includes the following source data and figure supplement(s) for figure 5:

**Source data 1.** Acquired pigmented images of the validation experiment.

**Source data 2.** Executed parameters and scores of the validation experiment.

**Source data 3.** ELISA scores.

**Source data 4.** Immunohistochemistry images.

**Source data 5.** Robot log.

**Figure supplement 1.** Video monitoring.

**Figure supplement 2.** Errors in the robotic operations.

the criteria for use in regenerative medicine research using the type of analysis measured in a previous clinical study (*Mandai et al., 2017*).

## Discussion

In this study, we proposed a robotic search system to autonomously search for optimal cell culture conditions, bringing together experimental robotics and BBO. Our robotic search system autonomously discovered the optimal combination of seven parameters comprising the iPSC-RPE induction process (target process) required to increase the number of pigmented cells (pigmentation score). Our approach can be applied to cell culture protocols other than iPSC-RPE induction; however, it may not be optimal even when implemented with a completely identical hardware-software setup. Below, we discuss some considerations and potential limitations for tailoring the components of our robotic search system (robots, parameters, and evaluation scores) to other targets.

Robots: the requirements depend on the nature of the target process. The search parameters must be changeable (flexibility), non-search parameters must remain stable or change only within the range of the specifications (reliability), and the operation must be sufficiently repeatable (accuracy). In addition, the storage capacity for $CO_2$ incubators and refrigerators needs to be set in accordance with the number of cell plates that are to be cultured concurrently. For target processes that require long-term culture (i.e. processes that have high retry costs) such as cell differentiation induction, the robots and peripheral equipment need to have low error rates. In target processes that have low retry costs, a lower priority on low error rates is required. We chose LabDroid for this research, as it meets these requirements and has good future operational extensibility.

Parameters: the number and range of searchable parameters is constrained by the number of experiments that can be performed. The more parameters to be searched, the greater the number of experiments required for sufficient optimization. The available experimental resources (number of iterations or parallel cultures) should be considered in advance for appropriate parameter optimization. Here, we limited the scope of our search to just seven parameters (*Table 1*). However, a myriad of potential parameter candidates, including other chemicals, culture media, and order of manipulations, can be considered. During parameter selection, we referred to previous cell culture studies and expert opinions, as well as preliminary simulations, to confirm that optimization was sufficiently feasible with our resources (*Figure 3—figure supplement 4*). The search ranges for the seven parameters were carefully selected for our target process; different appropriate search ranges should be selected in case of other target processes, including the induction of differentiation into other types of tissues.

Evaluation scores: since the optimization is performed on the evaluation scores, designing the evaluation function is critical. Here, we used the pigmentation score as the evaluation score because of the following reasons: when preparing iPSC-RPE cells for transplantation in clinical research, a clinical team evaluates the rate of pigmented cells, gene expression, and secretory substances in cells subjected to differentiation induction followed by purification (*Figure 1—figure supplement 1A*). This quality assessment is not based on a total score, and only those cells that satisfy all the criteria in all items are suitable for transplantation (*Kuroda et al., 2019*; *Mandai et al., 2017*; *Regent et al., 2019*). Because cell pigmentation is one of the criteria for the assessment, cell pigmentation alone is not sufficient to determine cell quality, but can be a requirement. It should be noted that the pigmentation score does not reflect the degree of pigmentation in individual cells, but indicates the number of cells in the dish whose pigmentation is above the threshold. Since pigmented cells and non-pigmented cells are mixed in the dishes at the end of the induction (i.e. before purification), single-cell omics analysis is needed to accurately evaluate the quality of individual cells. For example, in stem cells, a value (stemness index) has been proposed to evaluate stemness from single-cell mRNA-seq information (*Gulati et al., 2020*). We believe that if a similar index for iPSC-RPE cells indicating cell quality from transcriptome data is established, this could replace the pigmentation score that we used, and would make the process we have developed even more ideal.

## Materials and methods

**Key resources table**

| Reagent type (species) or resource | Designation | Source or reference | Identifiers | Additional information |
|---|---|---|---|---|
| Cell line (*Homo-sapiens*) | hiPSC 253G1 | RIKEN BRC | HPS0002 | |
| Antibody | Anti-ZO-1 (Rabbit polyclonal) | Thermo Fisher Scientific Inc. | 61–7300 | IHC (1:500) |
| Antibody | Anti-MITF (Mouse monoclonal) | Abcam plc. | ab80651 | IHC (1:1000) |
| Antibody | Alexa Fluor 488 Goat Anti-rabbit IgG (Goat polyclonal) | Thermo Fisher Scientific Inc. | A-11034 | IHC (1:1000) |
| Antibody | Alexa Fluor 546 Goat Anti-mouse IgG (Goat polyclonal) | Thermo Fisher Scientific Inc. | A-11030 | IHC (1:1000) |
| Sequence-based reagent | BEST1 (+) | This paper | RT-PCR primers | TAGAACCATCAGCGCCGTC |
| Sequence-based reagent | BEST1 (−) | This paper | RT-PCR primers | TGAGTGTAGTGTGTATGTTGG |
| Sequence-based reagent | RPE65 (+) | This paper | RT-PCR primers | TCCCCAATACAACTGCCACT |
| Sequence-based reagent | RPE65 (−) | This paper | RT-PCR primers | CCTTGGCATTCAGAATCAGG |
| Sequence-based reagent | CRALBP (+) | This paper | RT-PCR primers | GAGGGTGCAAGAGAAGGACA |
| Sequence-based reagent | CRALBP (−) | This paper | RT-PCR primers | TGCAGAAGCCATTGATTTGA |
| Sequence-based reagent | GAPDH (+) | This paper | RT-PCR primers | ACCACAGTCCATGCCATCAC |
| Sequence-based reagent | GAPDH (−) | This paper | RT-PCR primers | TCCACCACCCTGTTGCTGTA |
| Sequence-based reagent | RNeasy Micro Kit | QIAGEN | 74004 | |
| Sequence-based reagent | SuperScript III | Thermo Fisher Scientific Inc. | 18080–044 | |

*Continued on next page*

*Continued*

| Reagent type (species) or resource | Designation | Source or reference | Identifiers | Additional information |
|---|---|---|---|---|
| Commercial assay or kit | VEGF Human ELISA Kit | Thermo Fisher Scientific Inc. | BMS277-2 | |
| Commercial assay or kit | PEDF Human ELISA Kit | BioVendor | RD191114200R | |
| Chemical compound, drug | PD 173074 | Merck & Co., Inc. | P2499-5MG | |
| Chemical compound, drug | CultureSure Y-27632 | FUJIFILM Wako Pure Chemical Corporation | 036–24023 | |
| Chemical compound, drug | SB 431542 hydrate | Merck & Co., Inc. | S4317-5MG | |
| Chemical compound, drug | CKI-7 dihydrochloride | Merck & Co., Inc. | C0742-5MG | |
| Software, algorithm | LabDroid_optimizer | This paper | | Available at our Github (see Data and code availability) |
| Other | StemFit AK02N | Ajinomoto Co., Inc. | AK02N | see Materials and Methods >Reagents |
| Other | knockOut serum replacement (KSR) | Thermo Fisher Scientific Inc. | 10828028 | see Materials and Methods >Reagents |
| Other | FBS | Nichirei Corporation | 12007C | see Materials and Methods >Reagents |

## Guidelines

All experiments that involved the use of human-derived samples were reviewed and approved by the institutional review board of the Institutional Committee of RIKEN Kobe Branch (#Kobe1 2019–05 (3)).

## Reagents

hiPSC maintenance medium: 80% StemFit Basal Solution A and 20% StemFit iPS Expansion Solution B (#AK02N, Ajinomoto Co., Inc, Japan).

RPE differentiation medium (20% KSR): 0.10 mM MEM non-essential amino acids solution (NEAA) (#11140050, Thermo Fisher Scientific Inc, MA, USA), 1.0 mM sodium pyruvate (#S8636, Merck & Co., Inc, NJ, USA), 19% knockOut serum replacement (KSR) (#10828028, Thermo Fisher Scientific Inc, MA, USA), 0.0007% 2-mercaptoethanol (#139–06861, FUJIFILM Wako Pure Chemical Corporation, Japan), 78 U/mL benzylpenicillin sodium, and 78 µg/mL streptomycin sulfate (#15140122, Thermo Fisher Scientific Inc, MA, USA). All diluted in GMEM (#11710035, Thermo Fisher Scientific Inc, MA, USA).

RPE differentiation medium (15% KSR): 0.10 mM MEM NEAA (#11140050, Thermo Fisher Scientific Inc, MA, USA), 0.99 mM sodium pyruvate (#S8636, Merck & Co., Inc, NJ, USA), 15% KSR (#10828028, Thermo Fisher Scientific Inc, MA, USA), 0.0007% 2-mercaptoethanol (#139–06861, FUJIFILM Wako Pure Chemical Corporation, Japan), 82 U/mL benzylpenicillin sodium, and 82 µg/mL streptomycin sulfate (#15140122, Thermo Fisher Scientific Inc, MA, USA). All diluted in GMEM (#11710035, Thermo Fisher Scientific Inc, MA, USA).

RPE differentiation medium (10% KSR): 0.094 mM MEM NEAA (#11140050, Thermo Fisher Scientific Inc, MA, USA), 0.94 mM sodium pyruvate (#S8636, Merck & Co., Inc, NJ, USA), 10% KSR (#10828028, Thermo Fisher Scientific Inc, MA, USA), 0.0007% 2-mercaptoethanol (#139–06861, FUJIFILM Wako Pure Chemical Corporation, Japan), 85 U/mL benzylpenicillin sodium, and 85 µg/mL streptomycin sulfate (#15140122, Thermo Fisher Scientific Inc, MA, USA). All diluted in GMEM (#11710035, Thermo Fisher Scientific Inc, MA, USA).

RPE maintenance medium: 29% Nutrient Mixture F-12 (#N6658, Merck & Co., Inc, NJ, USA), 1.9 mM L-glutamine (#G7513, Merck & Co., Inc, NJ, USA), 1.9% B-27 supplement, serum free (#17504044, Thermo Fisher Scientific Inc, MA, USA), 96 U/mL benzylpenicillin sodium, and 96 µg/mL streptomycin sulfate (#15140122, Thermo Fisher Scientific Inc, MA, USA). All diluted in DMEM (Low glucose) (#D6046, Merck & Co., Inc, NJ, USA).

FGF receptor inhibitor (FGFRi) stock: PD 173074 (#P2499-5MG, Merck & Co., Inc, NJ, USA) diluted in DMSO (#D2650–5X5ML, Merck & Co., Inc, NJ, USA).

Rho-kinase inhibitor (Y) stock (8–10 mM): CultureSure Y-27632 (#036–24023, FUJIFILM Wako Pure Chemical Corporation, Japan) diluted in distilled water (Otsuka Pharmaceutical Factory, Japan) to a final 10 µM concentration when added to the cell culture medium.

TGF-β/Activin/Nodal signal inhibitor (SB) stock (4–5 mM): SB 431542 hydrate (#S4317-5MG, Merck & Co., Inc, NJ, USA) diluted in DMSO (#D2650−5X5ML, Merck & Co., Inc, NJ, USA) to a final 5 µM concentration when added to the cell culture medium.

Wnt signal inhibitor (CKI) stock (2.4–3 mM): CKI-7 dihydrochloride (#C0742-5MG, Merck & Co., Inc, NJ, USA) diluted in distilled water (Otsuka Pharmaceutical Factory, Japan) to a final 3 µM concentration when added to the cell culture medium.

RPE adhesion medium: DMEM/F12 (D8437, Merck & Co., Inc, NJ, USA), 10% FBS (12,007C, Nichirei Corporation, Japan).

RPE washing solution: 98% DMEM/F12 (D8437, Merck & Co., Inc, NJ, USA), 1 mM sodium pyruvate (S8636, Merck & Co., Inc, NJ, USA), 2 mM L-glutamine (G7513, Merck & Co., Inc, NJ, USA).

## Labware

For human use: micropipette tip, 2140-05-HR/2149P-05/61849, Thermo Fisher Scientific Inc (MA, USA); micropipette tip, 30389165, Mettler Toledo (OH, USA); micropipette tip, 737251, Greiner Bio-One International GmbH (Germany); disposable pipette, 356507, Corning Incorporated (NY, USA); disposable pipette, 606160/607160/760160/768160, Greiner Bio-One International GmbH (Germany); filtration, SLGVJ13SL, Merck & Co., Inc (NJ, USA); filtration, SS-10LZ, Terumo Corporation (Japan); filtration, 431096/430281/431097/430282, Corning Incorporated (NY, USA); 1.5 mL tube, 72.692MS, Sarstedt K.K. (Japan); 15 mL tube, 352096, Corning Incorporated (NY, USA); 50 mL tube, 352070, Corning Incorporated (NY, USA).

For LabDroid use: 6-well plate, 353046, Corning Incorporated (NY, USA); 50 mL tube, MS-58500, Sumitomo Bakelite Co., Ltd. (Japan); micropipette tip, 3511-05-HR/3512-05-HR/94410313/94410713 /94052550, Thermo Fisher Scientific Inc (MA, USA).

## LabDroid Maholo booth

LabDroid including peripheral equipment were placed inside a booth made of acrylic walls and a stainless steel frame with three fan-filter-units (*Figure 2—figure supplement 1*). The LabDroid booth included a dual-arm humanoid (Robotic Biology Institute Inc, Japan), a $CO_2$ incubator (APC-30D, ASTEC Co., Ltd., Japan), micropipettes (4641110N/4641030N/4641230N/4641210N, Thermo Fisher Scientific Inc, MA, USA), a tube rack (Robotic Biology Institute Inc, Japan), a plate rack (Robotic Biology Institute Inc, Japan), a dry bath (EC-40RA, AS ONE Corporation, Japan), a tip sensor (Robotic Biology Institute Inc, Japan), an aspirator (SP-30, Air Liquide, Italy), a dust bin (EPD3S, Sekisui Techno Moulding Co., Ltd., Japan), and a microscope (EVOS FL Auto 2, Thermo Fisher Scientific Inc, MA, USA).

## hiPSC culture — initiation and preparation of cell suspensions (human part)

The hiPSC line 253G1 (*Nakagawa et al., 2008*), made from human dermal fibroblasts, was obtained from RIKEN BRC (HPS0002). The hiPSCs were cultured and differentiated using the method previously described (*Haruta et al., 2004*; *Kawasaki et al., 2002*; *Osakada et al., 2008*). Mycoplasma contamination tests were performed periodically during the study and the results were always negative.

On DDay −14, frozen hiPSCs were initiated using the following procedures: first, laminin-coated 6-well plates were prepared. A final concentration of 0.5 µg/cm$^2$ iMatrix-511 (Matrixome Inc, Japan) diluted in PBS (-) was then added to each well of the four 6-well plates and incubated for a minimum of 60 min at 37 °C and 5% $CO_2$, after which 0.75 mL/well of hiPSC maintenance medium was added. The supernatant was then removed. Next, 1 mL/well of hiPSC maintenance medium containing Rho-kinase inhibitor (final 10 µM concentration) was added, and the coated plates were incubated at 37 °C and 5% $CO_2$ until further use.

For hiPSC initiation, frozen vials of hiPSCs stored in liquid nitrogen were thawed in a water bath set at 37 °C, and the cells were subsequently suspended in 5 mL of hiPSC maintenance medium. After centrifugation (160×*g*, 22 °C, 4 min), the supernatant was removed and an appropriate volume of hiPSC maintenance medium with a final 10 µM Rho-kinase inhibitor concentration was added. After

counting the cells with a hemocytometer, the cells were seeded into laminin-coated 6-well plates at 43,300–45,000 cells/1.5 mL medium/well.

On DDay −13, the medium was replaced with hiPSC maintenance medium without Rho-kinase inhibitor. On DDays −12 to −8, the medium was replaced with the same medium composition at 24–72 hr intervals. On DDay −7, cells were collected from the plate, and cell suspensions were delivered to the LabDroid booth. The medium was aspirated and 2 mL/well of PBS (-) was gently added and then aspirated for washing, followed by addition of 1 mL of 0.5 x TrypLE Select CTS (#A12859-01, Thermo Fisher Scientific Inc, MA, USA) diluted in 0.5 mM EDTA/PBS (-) and incubated at 37 °C and 5% $CO_2$ for 10–20 min. Then, cells were detached by pipetting and collected into a 50 mL tube, to which 1 mL of hiPSC maintenance medium and 3 mL of PBS (-) were added. After centrifugation (160×$g$, 22 °C, 4 min), the supernatant was removed, 0.75 mL of hiPSC maintenance medium with 10 µM Rho-kinase inhibitor was added, and the cells were resuspended. The cell suspension was filtered through a 40 µm cell strainer (#352340, Corning Incorporated, USA) with an additional 0.75 mL of hiPSC maintenance medium. After counting the cells with a hemocytometer, the cell suspension was set to 133,400 cells/20 mL with hiPSC maintenance medium containing 10 µM Rho-kinase inhibitor in eight 50 mL tubes. To prepare the cell suspensions, eight 6-well plates coated with laminin were prepared. A final concentration of 0.5 µg/cm$^2$ of iMatrix-511 (Matrixome Inc, Japan) diluted in PBS (-) was added to each well of four 6-well plates and incubated for a minimum of 60 min at 37 °C and 5% $CO_2$.

## iPSC-RPE differentiation (LabDroid part)

On DDay −7, the hiPSC suspension was seeded into eight 6-well plates by coating eight 6-well plates with laminin, and placing eight tubes of the iPSC suspension and labware in the appropriate positions. The task of seeding was initiated, and the robotic operation was performed by LabDroid (*Figure 2— figure supplements 2A and 3*; *Figure 4—video 1*). After the robotic operation, the eight cell-seeded plates were exported and incubated in a $CO_2$ incubator outside the LabDroid booth.

On DDay −6, the eight seeded plates were imported into the $CO_2$ incubator of the LabDroid booth. The users prepared eight 50 mL tubes of hiPSC maintenance medium with a final 10 µM Rho-kinase inhibitor concentration and two 50 mL tubes of hiPSC maintenance medium with final 5 µM FGFRi and 10 µM Rho-kinase inhibitor concentrations. The reagents and labware were placed in the appropriate positions. The task of preconditioning was then initiated, and the robotic operation was performed by LabDroid (medium exchange type I; *Figure 2—figure supplements 2B and 4*; *Figure 4—video 2*).

On DDays −5 to −1, the users prepared eight 50 mL tubes of hiPSC maintenance medium without Rho-kinase inhibitor and two 50 mL tubes of hiPSC maintenance medium with a final 5 µM FGFRi concentration. The reagents and labware were placed in the appropriate positions. The task of preconditioning was initiated, and the robotic operation was performed by LabDroid (medium exchange type I; *Figure 2—figure supplements 2B and 4*; *Figure 4—video 2*).

On DDay 0, the following procedure was used for the operation of four plates: the users prepared four 6-well plates coated with laminin. A final 0.5 µg/cm$^2$ concentration of iMatrix-511 (Matrixome Inc, Japan) diluted in PBS (-) was added to each well of the four 6-well plates and then the plates were incubated for a minimum of 60 min at 37 °C and 5% $CO_2$. The users also prepared two 50 mL tubes of PBS (-), two 50 mL tubes of 0.5 x TrypLE Select CTS (#A12859-01, Thermo Fisher Scientific Inc, MA, USA) diluted in 0.5 mM EDTA/PBS (-), and four plates with RPE differentiation medium (20% KSR) with final 10 µM Rho-kinase inhibitor/3 µM Wnt signal inhibitor/5 µM TGF-β/Activin/Nodal signal inhibitor (4 mL/well each) concentration. The cell plates, laminin-coated plates, plates with medium, reagents, and labware were placed in the appropriate positions. The task of passage was initiated, and robotic operations were performed by LabDroid (*Figure 2—figure supplements 2D and 5*; *Figure 4—video 3*). After performing this operation twice (four plates each), the eight cell-passaged plates were exported and incubated in a $CO_2$ incubator outside the LabDroid booth.

On DDay 1, the eight cell-passaged plates were imported into the $CO_2$ incubator of the LabDroid booth. Users prepared eight 50 mL tubes of RPE differentiation medium (10% KSR), two 50 mL tubes of 100% KSR, one 50 mL tube of 4 mM Rho-kinase inhibitor stock/1.2 mM Wnt signal inhibitor stock, and one 50 mL tube of 4 mM TGF-β/Activin/Nodal signal inhibitor stock. The reagents and labware were placed in the appropriate positions. The task of RPE differentiation was initiated, and the robotic operation was performed by LabDroid (medium exchange type I; *Figure 2—figure supplements 2B and 6*; *Figure 4—video 4*).

On DDays 2–19, the users prepared eight 50 mL tubes of RPE differentiation medium (10% KSR), two 50 mL tubes of 100% KSR, one 50 mL tube of 4 mM Rho-kinase inhibitor stock/1.2 mM Wnt signal inhibitor, and one 50 mL tube of 4 mM TGF-β/Activin/Nodal signal inhibitor. The reagents and labware were placed in the appropriate positions. The task of RPE differentiation was initiated, and the robotic operation was performed by LabDroid (medium exchange type I; *Figure 2—figure supplements 2B and 6*; *Figure 4—video 4*).

On DDays 20–32, the users prepared eight 50 mL tubes of RPE differentiation medium (10% KSR; DDays 10–25) or RPE maintenance medium (DDays 26–32). The reagents and labware were placed in the appropriate positions. RPE differentiation and maintenance were initiated and the robotic operations were performed by LabDroid (medium exchange type II; *Figure 2—figure supplements 2C and 7*; *Figure 4—video 5*).

## Scoring — sampling (human part)

On DDay 33, the cell plates were exported and the cell culture medium was replaced with fresh RPE maintenance medium. After 24 hr (DDay 34), the medium was collected for ELISA analysis. The remaining media were aspirated and 2 mL of PBS (-) were added and then aspirated for washing. After that, photographic images were acquired for the calculation of scoring values.

## Scoring — image analysis (human part)

Images were acquired using a digital camera (PSG7X MARKII, Canon Inc, Japan): ISO 500; focal length $F$=9.00, 50 mm; exposure time, 1/1250 s. The camera was set in the same position throughout all experiments. The acquired images were automatically processed by filtering with Gaussian blur, subtracting the background, binarizing by thresholding with a constant value, and cropping with a constant pixel value. The colored cell area was then calculated (*Figure 2—figure supplement 8*).

## Purification and storage (human part)

Purification of iPSC-RPE cells was conducted using the same protocol described in a study previously reported (*Mandai et al., 2017*). When the RPE colonies reached an appropriate size, the cells were suspended in RPE maintenance medium and kept as a floating culture for about 10 days in a low cell adhesion plate (MS-90600Z, Sumitomo Bakelite Co., Ltd., Japan). Under the microscope, colonies consisting only of black RPE cells were selected. Then, they were transferred to 12-well plates coated with iMatrix, and cultured in RPE adhesion medium/RPE maintenance medium (1:1). Once the RPE cell colonies became attached to the dish, they were cultured in RPE maintenance medium with basic fibroblast growth factor (bFGF), which was changed every 2–3 days.

After 10–12 days of cell selection, unsuitable cells were removed, and the cells were passaged. The medium was aspirated and 1 mL of RPE washing solution was added and aspirated again for washing. Then, 0.5 mL of RPE washing solution was added and atypical cells were eliminated using micropipette tips under microscope observation. After the removal process, the medium was aspirated, 1 mL/well of PBS (-) was added and aspirated for washing, and then 0.5 mL of Trypsin-EDTA solution (203–20251, FUJIFILM Wako Pure Chemical Corporation, Japan) was added, followed by incubation at RT and 5% $CO_2$ for 8–10 min. Cells were detached by pipetting and collected into a 50 mL tube. After centrifugation (280×$g$, 25 °C, 4 min), the supernatant was removed, and the pellet was resuspended in 1 mL/plate of RPE adhesion medium/RPE maintenance medium (1:1) and filtered through a 40 μm cell strainer (352340, Corning Incorporated, NY, U.S.A.). After counting the cells with a hemocytometer, the cells were seeded into 12-well plates. The medium was changed to RPE maintenance medium with bFGF.

After 1–3 days of cell passage, the medium was aspirated, the cells were washed with 0.5 mL of RPE maintenance medium, and 1 mL of RPE maintenance medium containing 10 ng/mL bFGF and 0.5 μM SB431542 was added. This medium was exchanged every 2–3 days.

The cells were stored when they formed hexagonal shapes after sufficient confluency. For that, the medium was aspirated, 1 mL/well of PBS (-) was added and then aspirated for washing, and 0.5 mL of Trypsin-EDTA solution (203–20251, FUJIFILM Wako Pure Chemical Corporation, Japan) was added, followed by incubation at 37 °C and 5% $CO_2$ for 10–15 min. After adding >0.5 mL of RPE adhesion medium, the cells were detached using a cell scraper (MS-93100, Sumitomo Bakelite Co., Ltd., Japan). The cell suspension was filtered through a 40 μm cell strainer (352340, Corning Incorporated, NY,

USA) and then centrifuged for 4 min at 280×*g* to obtain a cell pellet. The pellet was resuspended in 1 mL of RPE adhesion medium/RPE maintenance medium (1:1) and filtered through a 40 µm cell strainer. After counting the cells with a hemocytometer, the cell suspension was centrifuged for 4 min at 280×*g* to obtain a cell pellet. Then, STEM-CELLBANKER (CB047, Zenoaq Resource Co., Ltd., Japan) was added until a cell concentration of 500,000 cells/0.5 mL/tube, and the cell suspensions were dispensed into cryovials. The cryotubes were placed in a cell freezing container at −80 °C for 3–24 hr, and then stored at −150 °C.

### Initiation of iPSC-RPE stock and recovery culture (human part)

Frozen vials of RPE cells were thawed in a 37 °C water bath and suspended in 4.5 mL of RPE adhesion medium. After centrifugation (280×*g*, 25 °C, 4 min), the supernatant was removed and RPE adhesion medium/RPE maintenance medium (1:1) was added. After counting the cells with a hemocytometer, the cells were seeded into 24-well plates (0.5 mL/well).

After 1–3 days of cell seeding, the medium was aspirated, the cells were washed with 0.25 mL of RPE maintenance medium, and 0.5 mL/well of RPE maintenance medium containing 10 ng/mL bFGF and 0.5 µM SB431542 was added. This same type of medium was exchanged every 2–3 days.

Two weeks after seeding, the RPE cells were passaged. Two weeks after cell passage, the RPE cells were used for cell biological validation processes (RT-PCR, ELISA, and immunohistochemistry).

### Validation — RT-PCR (human part)

Total RNA was extracted from transfected cells using RNeasy Micro Kit (#74004, QIAGEN, Germany). First-strand cDNA synthesis was performed on 500–1000 ng of total RNA, using SuperScript III (#18080–044, Thermo Fisher Scientific Inc, MA, USA) according to the manufacturer's instructions. Each mRNA transcript was amplified using PCR with the following primers:

*BEST1* (+), 5'-dTAGAACCATCAGCGCCGTC
*BEST1* (−), 5'-dTGAGTGTAGTGTGTATGTTGG
*RPE65* (+), 5'-dTCCCCAATACAACTGCCACT
*RPE65* (−), 5'-dCCTTGGCATTCAGAATCAGG
*CRALBP* (+), 5'-dGAGGGTGCAAGAGAAGGACA
*CRALBP* (−), 5'-dTGCAGAAGCCATTGATTTGA
*GAPDH* (+), 5'-dACCACAGTCCATGCCATCAC
*GAPDH* (−), 5'-dTCCACCACCCTGTTGCTGTA

### Validation — ELISA (human part)

The collected media were centrifuged (90×*g*, 4 °C, 1 min), and the supernatant was collected and stored at −80 °C. The amount of VEGF contained in the thawed medium was measured using the protocols and reagents from the VEGF Human ELISA Kit (BMS277-2, Thermo Fisher Scientific, USA), and the amounts of PEDF were measured using a Human ELISA Kit (RD191114200R, BioVendor, Czech Republic).

### Validation — Immunohistochemistry (human part)

Cells were washed with PBS (-), fixed in 15% paraformaldehyde for 1 hr at RT (approximately 25 °C), and stored at 4 °C after removal of PFA and addition of PBS (-). After removal of the solutions, cells were treated with 50 µL/well of 0.2% Triton X-100/PBS (-), incubated for 30 min at RT, washed with PBS (-), blocked with 50 µL of Blocking One (03953–95, Nacalai Tesque Inc, Japan), and incubated for 1 h at RT. After removal of the solutions, cells were stained at 4 °C o/n in 50 µL of the 1st antibody diluent (rabbit anti-ZO-1, 61–7300, Thermo Fisher Scientific Inc, MA, USA; anti-MITF, mouse anti-MiTF, ab80651, Abcam plc., Britain; antibody diluent, S2022, Agilent Technologies Inc, USA). After removal of the solutions, cells were washed with PBS (-) and then stained at RT for 1 hr in 50 µL of the 2nd antibody diluent (Alexa Fluor 546 Goat Anti-mouse IgG, A-11030, Thermo Fisher Scientific Inc, MA, USA; Alexa Fluor 488 Goat Anti-rabbit IgG, A-11034, Thermo Fisher Scientific Inc, MA, USA; antibody diluent, S2022, Agilent Technologies Inc, USA) with DAPI (1 µg/mL, D1206, Thermo Fisher Scientific Inc, MA, USA). After removal of the solutions, cells were washed with PBS (-), and then 50 µL of PBS (-) was added. Images of immunohistochemistry samples were acquired using an IX73 inverted microscope (Olympus, Japan).

## Bayesian optimization module

When no prior experimental results exist, the Bayesian optimization module generates the next query from random uniform sampling. When past experimental results are available, the Bayesian optimization module generates queries using two components: the Model updater and the Query generator (*Figure 3C*).

The Model updater updates the surrogate model to predict the experimental results given past experimental results: $\mathcal{D} = \{(\boldsymbol{x}_i, y_i)\}_{i=1}^n$. We adopted Gaussian process regression (GPR, *Figure 3—figure supplement 1*) with the ARD-RBF kernel as the surrogate model to estimate the expected score and confidence level for all unevaluated experimental parameters. Based on the experimental results shown in *Figure 2E*, the observation noise was assumed to follow a zero-mean Gaussian noise with a variance of 0.0039 at all points in the search space. By using the surrogate model, the Query generator generates the next queries in two steps. In step 1, the Query generator constructs an acquisition function that estimates the expected progress toward the optimal experimental parameter at a given experimental parameter $x$ in the search space. We adopted the Expected improvement (EI) (*Jones et al., 1998*), a commonly used acquisition function in BO. EI estimates how much improvement over the current best score is expected from each point in the search space. In step 2, by using the acquisition function, the Query generator decides where to evaluate next, and our problem required the simultaneous performance of 48 experiments corresponding to 8 plates x 6 wells in each round. In addition, because the *DP* is a batch contextual parameter as described herein, a policy function that generates parameter sets taking such structural context into account must be incorporated. Therefore, we developed the Batch Contextual Local Penalization (BCLP) as a policy function to generate multiple points with context in parallel. The BCLP is a batch generation policy that extends the local penalization (*Gonzalez et al., 2016*) to be applied to cases where complex structural context parameters exist. As shown in *Figure 3E*, for each value of the contextual parameter *DP* in ascending order, BCLP iteratively generated the parameter by maximizing and penalizing the acquisition function 48 times to obtain the next experimental parameters $X_{\text{next}}$ for each subsequent well (Algorithm 1, 2). In addition, after each round, the more promising *KP* intervals were reconfigured by calculating the integral value of the acquisition function (Algorithm 3). We also replaced the queries that corresponded to the place of the top two pigmentation scores in the previous experiments with the parameter of the top two pigmentation scores in the previous experiments as a positive control. For more information about the optimization module, see the **Appendix**.

## Statistical analysis

Statistical analyses were performed by Wolfram Mathematica version 11.2.0.0. In this study, p<0.05 was considered significant (*p<0.05, **p<0.01, ***p<0.001, and n.s.=not significant).

## Data and code availability

All code that supports the findings of this study is available at https://github.com/labauto/LabDroid_optimizer, (copy archived at swh:1:rev:661ef792d4b7568a2e673178d9f1e6ed3c84ab1b, *Tsuzuki, 2022*). This code is based on GPyOpt (*GPyOpt: Gaussian Process Optimization using GPy*).

# Acknowledgements

We thank T Iwata, T Mitsuyama, H Ozaki, and N Yachie for their insightful comments; H Hirabayashi and E Takagi for research support; A Kato and H Uchida for illustrations; and J Freeman for carefully proofreading the manuscript. We also thank all the laboratory members at RIKEN BDR, in particular, N Koide, Y Shibata, A Maeda, and T Maeda for their kind help in the preparation of the materials, their support in the experiments, and their insightful discussions.

# Additional information

### Competing interests

Genki N Kanda: A stakeholder of Robotic Biology Institute, Inc. Taku Tsuzuki, Chihaya T Watanabe, Tatsuki Higashi, Shuhei A Horiguchi: An employee of Epistra Inc. Motoki Terada, Noriko Sakai,

Tomohiro Masuda, Mitsuhiro Nishida: An employee of VCCT Inc. Taku Kudo, Motohisa Kamei: An employee of Robotic Biology Institute, Inc. Kenji Matsukuma, Tohru Natsume: An executive of Robotic Biology Institute, Inc. Takeshi Sakurada, Yosuke Ozawa: An executive of Epistra Inc. Masayo Takahashi: An executive of Vision Care Inc and VCCT Inc. Koichi Takahashi: A shareholder of Epistra Inc. The other authors declare that no competing interests exist.

## Funding

| Funder | Grant reference number | Author |
|---|---|---|
| Japan Agency for Medical Research and Development | JP20bm0204002 | Masayo Takahashi |
| New Energy and Industrial Technology Development Organization | | Masayo Takahashi |
| Japan Science and Technology Agency | JPMJMI18G4 | Koichi Takahashi |
| Japan Science and Technology Agency | JPMJMI20G7 | Koichi Takahashi |
| RIKEN | Engineering Network | Masayo Takahashi |
| RIKEN | Junior Research Associate Program | Naohiro Motozawa |

The funders had no role in study design, data collection and interpretation, or the decision to submit the work for publication.

## Author contributions

Genki N Kanda, Investigation, Methodology, Project administration, Visualization, Writing – original draft, Writing – review and editing; Taku Tsuzuki, Formal analysis, Methodology, Software, Validation, Visualization, Writing – original draft, Writing – review and editing; Motoki Terada, Noriko Sakai, Naohiro Motozawa, Tomohiro Masuda, Mitsuhiro Nishida, Genshiro A Sunagawa, Investigation, Writing – review and editing; Chihaya T Watanabe, Tatsuki Higashi, Shuhei A Horiguchi, Software, Writing – review and editing; Taku Kudo, Motohisa Kamei, Resources, Writing – review and editing; Kenji Matsukuma, Funding acquisition, Resources, Writing – review and editing; Takeshi Sakurada, Project administration, Software, Writing – review and editing; Yosuke Ozawa, Funding acquisition, Software, Writing – original draft, Writing – review and editing; Masayo Takahashi, Funding acquisition, Resources, Supervision, Writing – review and editing; Koichi Takahashi, Conceptualization, Funding acquisition, Writing – review and editing, Supervision, Writing – original draft; Tohru Natsume, Conceptualization, Funding acquisition, Supervision, Writing – review and editing

## Author ORCIDs

Genki N Kanda (iD) http://orcid.org/0000-0002-6372-241X
Shuhei A Horiguchi (iD) http://orcid.org/0000-0002-8459-1914
Tohru Natsume (iD) http://orcid.org/0000-0002-1510-2582

## Ethics

Human subjects: All experiments that involved the use of human-derived samples were reviewed and approved by the institutional review board of the Institutional Committee of RIKEN Kobe Branch (#Kobe1 2019-05 (3)).

## Decision letter and Author response

Decision letter https://doi.org/10.7554/eLife.77007.sa1
Author response https://doi.org/10.7554/eLife.77007.sa2

---

## Additional files

### Supplementary files
• Transparent reporting form

## Data availability

All data generated or analysed during this study are included in the manuscript and supporting file. All code that supports the findings of this study is available at https://github.com/labauto/LabDroid_optimizer (copy archived at swh:1:rev:661ef792d4b7568a2e673178d9f1e6ed3c84ab1b).

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

## Appendix 1

### 1. Details about parameterization of the iPSC-RPE protocol

The addition of FGFRi to the medium during preconditioning significantly increased the efficiency of induction of differentiation into RPE cells (*Kuroda et al., 2019*). The unit of *PC* is nM, and it takes values from 0 to 505, while *PP* is the number of days FGFRi is added, taking values from 1 to 6. For example, *PC*=300 and *PP*=2 indicates that 300 nM of FGFRi is added to DDays –2 and –1, and not from DDays –6 to –3.

*DS* was adopted because shear stress affects cell properties (*Wall and Davie, 2013*), and *DL* was adopted because the number of cells at passage is empirically known to contribute to the efficiency of differentiation induction. *DS* takes integer values from 10 to 100, indicating the speed at which the micropipette syringe is pressed (unit: mm/sec); *DL* takes two integer values, one long and one short, indicating the bottom surface area to be pipetted. We adopted *DP* at passage because of the constraints in the configuration of the LabDroid, which required different fixed values. *DP* takes integer values of 5, 8, 11, 14, 17, 20, and 23 to indicate the number of minutes that the trypsin solution was added and incubated at room temperature after incubation at 37 °C for 20 min.

The three chemical supplements are Rho-kinase inhibitor, TGF-β/Activin/Nodal signal inhibitor, and Wnt signal inhibitor, all of which are known to affect the culture and differentiation of iPS cells (*Osakada et al., 2009*; *Watanabe et al., 2007*). KSR was added to the medium at a 20% concentration on DDay 0 and was progressively reduced to 10%. Under baseline conditions, KSR is added in the following manner: 20% on DDays 0–4, 15% on DDays 5–7, and 10% from DDay 8, suggesting that this rate of reduction is important for the induction of RPE differentiation (*Kuroda et al., 2019*). The *KP* takes values from 1 to 19 and the concentration of KSR is decreased linearly daily until KSR becomes 10% on the DDay of the *KP* value. For example, when *KP*=4, KSR will be 20% on DDay 0, 17.5% on DDay 1, 15.0% on DDay 2, 12.5% on DDay 3, and 10% on DDay 4. *3P* takes integer values from 3 to 19, and indicates the day in which the three chemical supplements are included in the differentiation-inducing medium from DDay 0 to the DDay of that *3P* value.

### 2. Gaussian process overview

In Bayesian optimization, the Gaussian process (GP) $p(f) = \mathcal{GP}(\mu, \sigma; k)$ is typically used as the surrogate model to approximate the I/O of the objective function $f$. The GP is characterized by three components: a mean function μ, a variance function $\sigma$ and a kernel function (positive-value covariance function) $k$. In case of past experimental data $\mathcal{D} = \{(\boldsymbol{x}_i, y_i)\}_{i=1}^{n}$ of $n$ observations, and Gaussian prior $\mathcal{GP}(0, \sigma; k)$, Gaussian Posterior is then given in $\mathcal{GP}(\mu_n, \sigma_n; k)$, where

$$\mu_n(\boldsymbol{x}) = \boldsymbol{k}_n(\boldsymbol{x})^T \left(\boldsymbol{K}_n + \sigma^2 \boldsymbol{I}\right)^{-1} \boldsymbol{y}_n$$

$$\sigma_n^2(\boldsymbol{x}) = k(\boldsymbol{x}, \boldsymbol{x}) - \boldsymbol{k}_n(\boldsymbol{x})^T \left(\boldsymbol{K}_n + \sigma^2 \boldsymbol{I}_n\right)^{-1} \boldsymbol{k}_n(\boldsymbol{x})$$

$\mu_n(\boldsymbol{x})$ is the mean of Gaussian process posterior function, and $\sigma_n^2(\boldsymbol{x})$ is the variance of Gaussian process posterior function. In this work, we assume that the kernel function $k$ is stationary. $\sigma^2$ is the observation noise of the experiment.

*Figure 3—figure supplement 1* shows an example of a Gaussian process prior to being updated to a Gaussian posterior using observations. The uncertainty of GP distribution decreases around the observed points and increases further away from observations. For a more comprehensive explanation of Gaussian process regression, see *Rasmussen and Williams, 2006*.

### 3. Bayesian optimization overview

Bayesian optimization is a common black-box optimization method used to determine the input parameter $x$, which maximizes the objective function $f$ with as few executions as possible using iterative experiment-observation loops.

$$\boldsymbol{x}_{\max} = \arg\max_{\boldsymbol{x} \in \mathcal{X}} f(\boldsymbol{x})$$

*Figure 3—figure supplement 2* shows a toy example of how sequential Bayesian optimization generates the next experimental parameter. Bayesian optimization iteratively generates the

following experimental parameters in three steps: construction of a surrogate model (e.g., Gaussian process regression) for past experimental results (left column), definition of an acquisition function $\alpha(\boldsymbol{x}; \mathcal{D})$ (e.g., expected Improvement), and optimization of the acquisition function to obtain the next experimental parameter (right column). For a more comprehensive explanation and application of Bayesian optimization, see *Frazier and Wang, 2015* and *Shahriari et al., 2016*.

## 4. Bayesian optimization for the iPSC-RPE protocol

In the batch Bayesian optimization for the iPSC-RPE differentiation protocol, we generated the next experimental queries for each round in four steps (Algorithm 1). First, we constructed a Gaussian process posterior using a past experimental dataset. When no prior experimental results existed, we generated the next query from random uniform sampling. Next, using the Gaussian process posterior, we obtained an acquisition function. In this study, we used the expected improvement (*Jones et al., 1998*) as the acquisition function. Third, using the acquisition function, we generated a next experimental parameter set $X_{\text{next}}$ using a policy function. In this study, we used batch contextual local penalization (BCLP) as the batch generation policy. Finally, after executing experiments on the parameter set $X_{\text{next}}$, we computed the optimal context for the Detach trypsin Period $x_{DP}$ for the next round.

In Gaussian process regression, we assume the ARD-RBF kernel function $k$ as follows:

$$k_{\text{ARD}-\text{RBF}}\left(\boldsymbol{x},\boldsymbol{x}';\boldsymbol{\theta}\right) = \exp\left(-\frac{1}{2}\sum_{m=1}^{d}\left(\frac{x-x'}{l_m}\right)^2\right)$$

$\boldsymbol{\theta} = \{l_m\}^d$ represents d-dimensional length scale hyper-parameters that were optimized using the maximum-likelihood estimation in every GP fitting.

Based on the experimental results in *Figure 2E*, the observation noise was assumed to follow a normal distribution with a variance of 0.039, at all points in the search space.

In this work, we used the expected improvement as an acquisition function. The expected improvement estimates how much improvement over the current best score $X_{\text{next}}$ is expected from each one of the input parameters $X_{\text{next}}$ in the search space, as shown in the following form:

$$EI(\boldsymbol{x};\mathcal{D}) = \left(y_{\max} - \mu_n(\boldsymbol{x})\right)\Phi(Z) + \sigma_n(\boldsymbol{x})\phi(Z), \quad Z = \frac{y_{\max} - \mu_n(\boldsymbol{x})}{\sigma_n(\boldsymbol{x})}$$

$\mathcal{D}$ is the set of the past experimental results. $\Phi$ is the standard normal cumulative distribution function, and $\phi$ is the standard normal probability density function.

In the policy function (see *Figure 3E*), BCLP iteratively generated the parameter for each value of the contextual parameter *DP* in ascending order by maximizing and penalizing the acquisition function 48 times to obtain the next experimental parameters $X_{\text{next}}$. Here we show the algorithm that generates $X_{\text{next}}$ (*Figure 3E*). Starting from plate1, well1, BCLP fixes the value of the *DP* corresponding to a given well number and then maximizes the acquisition function to generate an experimental parameter for the well. The penalization of the acquisition function is then performed using a hammer function to the point where the experimental parameter is generated. Similarly, for the next well, an experimental parameter is generated by maximizing the penalized acquisition function and is penalized using the hammer function. By repeating such maximization-penalization loops, BCLP generates the next experimental parameters for each subsequent well.

In Algorithm 2, the function $\varphi(\boldsymbol{x}; \boldsymbol{x}_j)$ is a local penalizer of $\alpha(\boldsymbol{x}; \mathcal{D})$ at $\boldsymbol{x}_j$ such that:

$$\varphi\left(\boldsymbol{x};\boldsymbol{x}_j, \hat{L}\right) = \frac{1}{2}erfc(-z)$$

where

$$z = \frac{1}{\sqrt{2\sigma_n^2(\boldsymbol{x}_j)}}\left(\hat{L}\left\|\boldsymbol{x}_j - \boldsymbol{x}\right\| - \hat{M} + \mu_n\left(\boldsymbol{x}_j\right)\right)$$

for *erfc* the complementary error function, $\hat{M} = \max_i\{y_i\}$ is the best score in the past experiments and $\hat{L} = \max_{\chi}\|\mu_{\nabla}(\boldsymbol{x})\|$ is an approximated Lipschitz constant. Both $\hat{M}$ and $\hat{L}$ were calculated in each round. We used $g(z) = z$ when the acquisition function $\alpha(\boldsymbol{x}; \mathcal{D})$ was positive and the $g(z) = softplus(z)$ elsewhere (*Gonzalez et al., 2016*).

For the contextual parameter $x_{DP}$, contexts on a fixed trypsin processing time $c_{DP,t,i}$ are assigned for each well depending on the round number $t$ and the well number , because of the implementation constraints of the protocol. After each round, the context on the $x_{DP}$ is reconfigured as a variable. In the initial state (round $t = 1$), trypsin treatment time was assigned to the smallest well number (well 1) for eight minutes, and then it increased by three minutes for each additional well number.

$$\boldsymbol{c}_{DP,1} := (c_{DP,1,1}, c_{DP,1,2}, c_{DP,1,3}, c_{DP,1,4}, c_{DP,1,5}, c_{DP,1,6}) = (8, 11, 14, 17, 20, 23)$$

In the next round, the context on $x_{DP}$ was allowed to move back and forth $\Delta c = 3$ in parallel while maintaining a three minute interval processing time between the different wells. Thus, the context in round 2 $\boldsymbol{c}_{DP,2}$ will be $\boldsymbol{c}_{DP}$ or $\boldsymbol{c}_{DP}^-$ or $\boldsymbol{c}_{DP}^+$ as follows:

$$\boldsymbol{c}_{DP} := \boldsymbol{c}_{DP,1} = (8, 11, 14, 17, 20, 23)$$

$$\boldsymbol{c}_{DP}^- := \boldsymbol{c}_{DP,1} - \Delta c = (5, 8, 11, 14, 17, 20)$$

$$\boldsymbol{c}_{DP}^+ := \boldsymbol{c}_{DP,1} + \Delta c = (11, 14, 17, 20, 23, 26)$$

Using the flow shown in Algorithm 3 in each round, we could choose whether the context in the next round $\boldsymbol{c}_{DP,t+1}$ will be $\boldsymbol{c}_{DP}$, $\boldsymbol{c}_{DP}^-$ or $\boldsymbol{c}_{DP}^+$, based on the past experimental data.

We performed a regression with GPR in each round and defined the acquisition function EI based on the regressions, deriving the integral value of the EI for $x_{DP}$ for each different candidate interval of $x_{DP}$ (either at 5 min, 8 min, or 11 min start), respectively. When any derived integrals were improved by 5% or more compared to the integrals in the interval used in the current round, the interval for the *DP* for the next round was run in the interval that showed the largest integral value among the candidate intervals.

## 5. Testing optimization in simulation

Bayesian optimization was tested by optimizing the seven dimensional toy testing function shown in *Figure 3—figure supplement 3* under different conditions. *Figure 3—figure supplement 4A* shows a comparison between the performance of batch Bayesian optimization and that of random search when the scale of the observation noise on the testing function was changed. The testing function with different noise (SD=0.000, 0.064, 0.400) was optimized using batch Bayesian optimization with batch contextual local penalization (BCLP) and uniform random sampling, using 8 plates × 6 wells=48 queries per round. In each condition, we performed 18 independent experiments. When the observation noise was not present (SD=0.000) or was sufficiently small (SD=0.064), as in the proposed system shown in *Figure 2E*, BCLP shows better convergence performance compared to the random search. When the scale of the observation noise was relatively large (SD=0.400), the performance of BCLP was only as good as that of the random search case.

*Figure 3—figure supplement 4B* compares the performances of BCLP when the batch size was changed. For the benchmark function (noise SD=0.064), optimization was performed using a different number of plates (Np) per round. In each condition, we performed n=18 independent experiments and showed that the convergence performance improved as the number of plates (the batch size) increased.

