## [Editor Report]

The manuscript by Kanda GN, Natsume T et al. describes a robotic artificial intelligence system with a batch Bayesian optimization algorithm that allows to optimise and reliably repeat cell culture protocols. The authors utilise induced pluripotent stem cell-derived retinal pigment epithelial cells as a model culture system of broad interest in regenerative medicine. They demonstrate that the robotic system with a Bayesian algorithm accelerates the optimisation of cell culture protocols and increases the quality and quantity of cell products, compared with manual operations – these results will likely inform and strongly impact modern cell culture strategies in regenerative medicine. The manuscript clearly explains the parameters analysed, the methods and analyses performed, current limitations and possible broader future use beyond the system tested.

---

## [Decision Letter]

**Decision letter after peer review:**

Thank you for submitting your article "Robotic Search for Optimal Cell Culture in Regenerative Medicine" for consideration by *eLife*, and apologies for the protracted review process. Your article has been positively reviewed by 3 peer reviewers, including Simón Méndez-Ferrer as the Reviewing Editor and Reviewer #1, and the evaluation has been overseen by a Reviewing Editor and Mone Zaidi as the Senior Editor. The following individuals involved in the review of your submission have agreed to reveal their identity: Sujith Sebastian (Reviewer #2); Christelle Monville (Reviewer #3).

Essential revisions:

1) Given the potentially strong impact of this study on future automation of cell culture processes, it would be helpful to rephrase some general sentences to more specifically describe the results (the relevance is appreciated in any case) and more precisely indicate the cell culture parameters tested/optimised in the current model, and those amenable to be subsequently tested in future applications. Similarly, a clear description of potential limitations (such as necessary manual optimisation or certain conditions or complex protocols less amenable to be automated) would be helpful in order to inform the readership on the potential implications of these results, beyond induced pluripotent stem cell-derived retinal pigment epithelial cells.

2) The authors take the simple "pigmentation" criteria as the optimisation's read-out. I would suggest describing a little more how they define the pigmentation score and how they judge that it is relevant to good cell quality. They have assessed the function and phenotype of the cells by the measure of classical markers (by qPCR, IF and Elisa) but they should correlate "pigmentation score" and those results.

Moreover, I would suggest discussing if this approach is robot-dependant or if it would be suitable for any type of culture automate.

---

## [Author Response]

Essential revisions:1) Given the potentially strong impact of this study on future automation of cell culture processes, it would be helpful to rephrase some general sentences to more specifically describe the results (the relevance is appreciated in any case) and more precisely indicate the cell culture parameters tested/optimised in the current model, and those amenable to be subsequently tested in future applications.

We agree with the reviewers that a more precise description of the elucidation of the parameters makes the manuscript more “self-contained.” In this study, we focused on the seven parameters shown in Table 1 and described in "Parameterization of the protocol" in the Results. The iPSC-RPE induction protocol (target process) has many other potential parameters, including other chemicals, culture media, and the order of manipulation. Due to computational combination limits and the practical limits on the number of physical attempts that can be performed by a robot, we searched for an appropriate set of parameters based on previous cell culture studies and expert opinions. The seven parameters are search ranges carefully selected for the current target process. Thus, different appropriate parameters should be selected as search ranges for other target processes, including induction of differentiation into other tissue types. We have explained this in the Discussion.

Similarly, a clear description of potential limitations (such as necessary manual optimisation or certain conditions or complex protocols less amenable to be automated) would be helpful in order to inform the readership on the potential implications of these results, beyond induced pluripotent stem cell-derived retinal pigment epithelial cells.

We have modified the Discussion to include this point.

2) The authors take the simple "pigmentation" criteria as the optimisation's read-out. I would suggest describing a little more how they define the pigmentation score and how they judge that it is relevant to good cell quality. They have assessed the function and phenotype of the cells by the measure of classical markers (by qPCR, IF and Elisa) but they should correlate "pigmentation score" and those results.

In this study, we demonstrated that robotic search can be applied to search conditions for transplanted cells by first screening with the pigmented cell rate (score) as an evaluation indicator and then evaluating other criteria (RT-PCR, ELISA, IHC) in purified cells according to previous studies (Kuroda et al., 2019; Mandai et al., 2017; Regent et al., 2019). When preparing iPSC-RPE cells for transplantation in clinical research, a clinical team evaluates the rate of pigmented cells, gene expression, and secretory substances in cells subjected to differentiation induction followed by purification. This quality assessment is not based on a total score, and only those that satisfy all the criteria in all items are suitable for transplantation. Because cell pigmentation is one of the criteria for the assessment, cell pigmentation is not a sufficient condition for cell quality, but can be a requirement. For this reason, rates of pigmented cells are usually used as an indicator of differentiation induction efficiency.

The pigmentation score used in this study does not reflect the degree of pigmentation in individual cells, but indicates how many cells in the dish have pigmentation above the threshold. It is important to note that, since pigmented and unpigmented cells are mixed in the dish at the time when induction ends (before the purification process), single-cell RNA-seq is required to correctly evaluate the correlation between pigmentation and internal state (transcriptome) in individual cells; its implementation is a future challenge for us. Even in the case of evaluating cell quality by single-cell RNA-seq, cell quality (iPSC-RPE-ness) must be expressed as a one-dimensional value to apply the BBO algorithm adopted in this paper. To our knowledge, there is currently no way of defining a single value as “iPSC-RPE-ness” from transcriptome data. Hence, the development of the method is out of the scope of this study. In the future, the development of an evaluation value of “iPSC-RPE-ness” will allow us to replace the cell pigmentation rate used as the score in this paper. We believe that this will further improve our developed process.

To provide appropriate information to the reader, we have added explanatory text in the Discussion.

Moreover, I would suggest discussing if this approach is robot-dependant or if it would be suitable for any type of culture automate.

Our approach is not robot-dependent. However, the results of many types of cell culture experiments can be affected by subtle changes in operations, and require careful design and calibration of the hardware. In this study, we chose LabDroid Maholo because of its flexibility, reliability, and accuracy, which are characteristics required for the iPSC-RPE induction process. Our architecture and computational algorithms should work well with other types of culture automation hardware (or even humans) if these exhibit sufficient accuracy and reliability. We emphasized this in the Discussion.